# Self-*Soup*ervision: Cooking Model Soups without Labels

**Anthony Fuller** [1 2]   **James R. Green** [1]   **Evan Shelhamer** [1 2 3]

## Abstract

Model soups are strange and strangely effective combinations of parameters. They take a model (the stock), fine-tune it into multiple models (the ingredients), and then mix their parameters back into one model (the soup) to improve predictions. While all known soups require supervised learning, and optimize the same loss on labeled data, our recipes for Self-*Soup*ervision generalize soups to self-supervised learning (SSL). Our Self-Souping lets us flavor ingredients on new data sources, e.g. from unlabeled data from a task for transfer or from a shift for robustness. We show that Self-Souping on corrupted test data, then fine-tuning back on uncorrupted train data, boosts robustness by +3.5% (ImageNet-C) and +7% (LAION-C). Self-*Soup*ervision also unlocks countless SSL algorithms to cook the diverse ingredients needed for more robust soups. We show for the first time that ingredients can differ in their SSL hyperparameters—and more surprisingly, in their SSL algorithms. We cook soups of MAE, MoCoV3, MMCR, and LeJEPA ingredients that are more accurate than any single SSL ingredient. Code is available: https://github.com/antofuller/self_soupervision

## 1. Introduction: More Soups, Less Supervision

Model soups make several models (the ingredients) by independent fine-tunings initialized from a single model (the stock), then merge them back into one model (the soup) by mixing parameters to improve prediction accuracy (Wortsman et al., 2022). Each fine-tuning varies in its configuration (e.g. optimization hyperparameters) and each mixing can be a simple average or more sophisticated linear combination. In this way, soups convert more training time into more ac-

[1]Carleton University, Ottawa, Canada [2]Vector Institute, Toronto, Canada [3]University of British Columbia, Vancouver, Canada. Correspondence to: Anthony Fuller .

*Proceedings of the 43rd International Conference on Machine Learning*, Seoul, South Korea. PMLR 306, 2026. Copyright 2026 by the author(s).

curacy without more inference time: the soup model needs only as much computation as the original model.

Model soups are surprisingly possible, in that mixing model parameters is absolutely not guaranteed to result in a better model (or even an equally good model!). They are also surprisingly productive with improvements across many settings: vision (Jain et al., 2023; Wortsman et al., 2022), language (Ablin et al., 2025; Jang et al., 2023; Chronopoulou et al., 2023), text-to-image (Biggs et al., 2024), federated learning (Chen et al., 2024), domain generalization (Ramé et al., 2023; Ramé et al., 2022), and class imbalance (Aminbeidokhti et al., 2025). However all known soups have to train ingredients by *supervised* learning—depriving our palettes of tasty new soups for many occasions.

**We thus introduce Self-Soups, which are model soups made from ingredients that differ in their independent *self-supervised* training runs.** Self-Souping vastly expands the menu of possible soups by harnessing different losses to flavor ingredients from different distributions without requiring labels (Fig. 1). We can "inter-train" models to make ingredients by self-supervised learning (SSL), after pre-training but before fine-tuning to a task, to enable transfer and robustness by optimizing more losses on more data.

Self-*Soup*ervision unlocks countless self-supervised losses for preparing the *diverse* ingredients that robust soups need. For example, in §4.2 we first promote ingredient diversity by inter-training using 4 fundamentally different self-supervised losses, which we then fine-tune with labels and mix for improved robustness. These ingredients differ in their SSL runs (for preparation) *and* supervised training runs (for specialization). We can also inter-train on the *test* distribution to make shift-aware ingredients that improve robustness. For example, in §4.3 we inter-train on corrupted data (ImageNet-C (Hendrycks & Dietterich, 2019) and LAION-C (Li et al., 2025)), then fine-tune back on the distribution for which labels are available (ImageNet training data (Russakovsky et al., 2015)) to boost accuracy by +3.5 and +7%. In §4.4 we show that Self-*Soup*ervision helps transfer pre-trained models to 21 diverse visual tasks (the VTAB collection (Zhai et al., 2020)). In our final experiment (§4.6), we mix ingredients that differ only in their SSL runs—made possible by Self-*Soup*ervision. For each VTAB dataset, we run self-supervised inter-trainings that

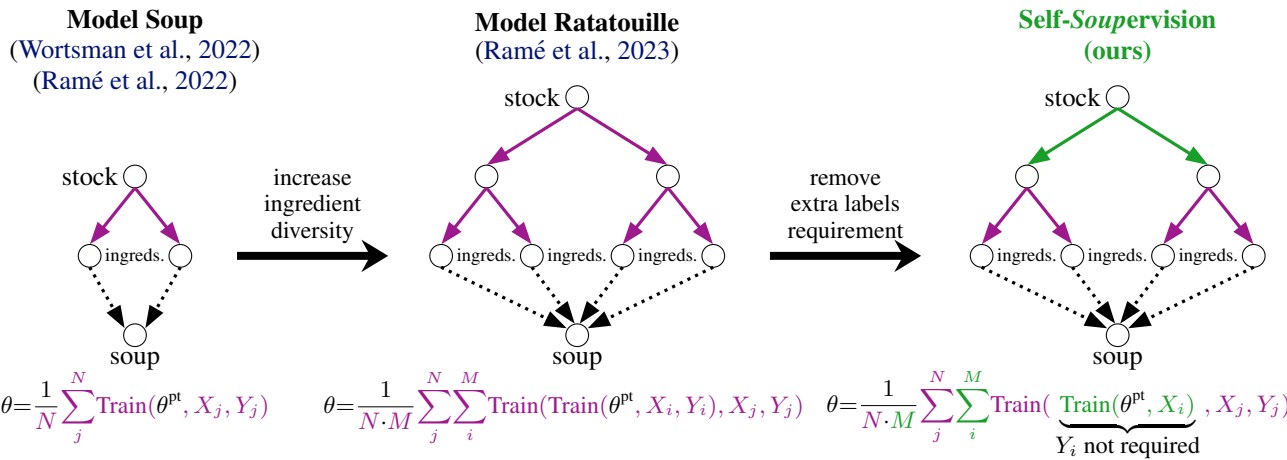

*Figure 1.* **Soups & Supervision.** Soups fine-tune then mix models to improve predictions. The original Model Soup (left) fine-tunes across hyperparameters for the task, while Model Ratatouilles (center) first inter-trains on *different labeled data* then fine-tunes for the task. Both are supervised and cannot harness unlabeled data. Our Self-*Soup*ervision (right) inter-trains across different losses and data to make more and better soups with and without labels. Our Self-Soups fine-tune then mix into a supervised model for the task. $\theta^{\text{pt}}$ is the pre-trained stock for initialization, $N$ is the number of fine-tunings on data $j$, $M$ is the number of inter-trainings on data $i$, and $\theta$ is the final model. We color **supervised** and **self-supervised** training runs / components. "ingreds." is short for model-soup ingredients.

differ in their self-supervised algorithms and algorithmic hyperparameters. We then mix these purely self-supervised ingredients by quickly "seasoning" (Croce et al., 2023) them: choosing the mixture conditioned on few-shot labels. We even mix soups for a task without training labels by a new and fully unsupervised variant that we call Self-Seasoning.

# 2. Background: Supervised Soups and Self-supervised Learning

## 2.1. Supervised Model Soups

**Initializing cooking with a stock.** Soups require that the models for mixing (the ingredients) share the same initial parameters for optimization (the stock).

**Adding ingredients by fine-tuning.** Each fine-tuning of the stock creates an ingredient for mixing a soup. Multiple *different* fine-tunings are key for different ingredients: soups rely on differences among the ingredient models for their gains. To vary their ingredients, Model Soups (Wortsman et al., 2022) vary the optimization parameters, such as the learning rate, data augmentation, and optimizer.

**Boosting ingredient diversity via inter-training.** Model Ratatouille (Ramé et al., 2023) fine-tunes in two stages, optimizing ingredients for longer, and increasing their diversity for domain generalization. Ratatouille first initializes with a stock, then "inter-trains" on up to 5 auxiliary labeled datasets independently, and finally fine-tunes these models on the target task for mixing. Ratatouille gains 0.5% when tested out of distribution. Although this gain is modest, like souping, Ratatouille produces a single model without

increased inference/deployment costs. Furthermore, Ratatouille showed that soups could be made from ingredients trained on different labeled datasets. We show soups can be made from ingredients trained on different unlabeled datasets and different self-supervised losses before mixing.

## 2.2. Linear Mode Connectivity

Not all models can be mixed. When mixing works, for ingredients that share a stock, the condition of linear mode connectivity holds. Linear mode connectivity (LMC) holds for two models if the accuracy of the interpolated model weights is greater than or equal to the interpolated accuracy of the models. Formally, for all $\lambda \in [0, 1]$,

$$\begin{aligned}\mathbf{acc}\left((1-\lambda) \cdot \theta_A + \lambda \cdot \theta_B\right) \geq \\ (1-\lambda) \cdot \mathbf{acc}(\theta_A) + \lambda \cdot \mathbf{acc}(\theta_B)\end{aligned} \tag{1}$$

where $\lambda$ is the interpolation weight, $\theta_A, \theta_B$ are the model weights, and **acc** is the accuracy.

Although model soups can provide gains, the known cases in which LMC holds are few. Our work adds another case. All methods initialize from a shared stock, but differ in how they optimize ingredients:

- Fine-tuning with *different stochasticity* (e.g. order, augmentation) (Frankle et al., 2020)
- Fine-tuning with *different hyper-parameters* (e.g. learning rate) (Wortsman et al., 2022)
- Inter-training on *different supervised datasets*, then fine-tuning on a target task (Ramé et al., 2023)

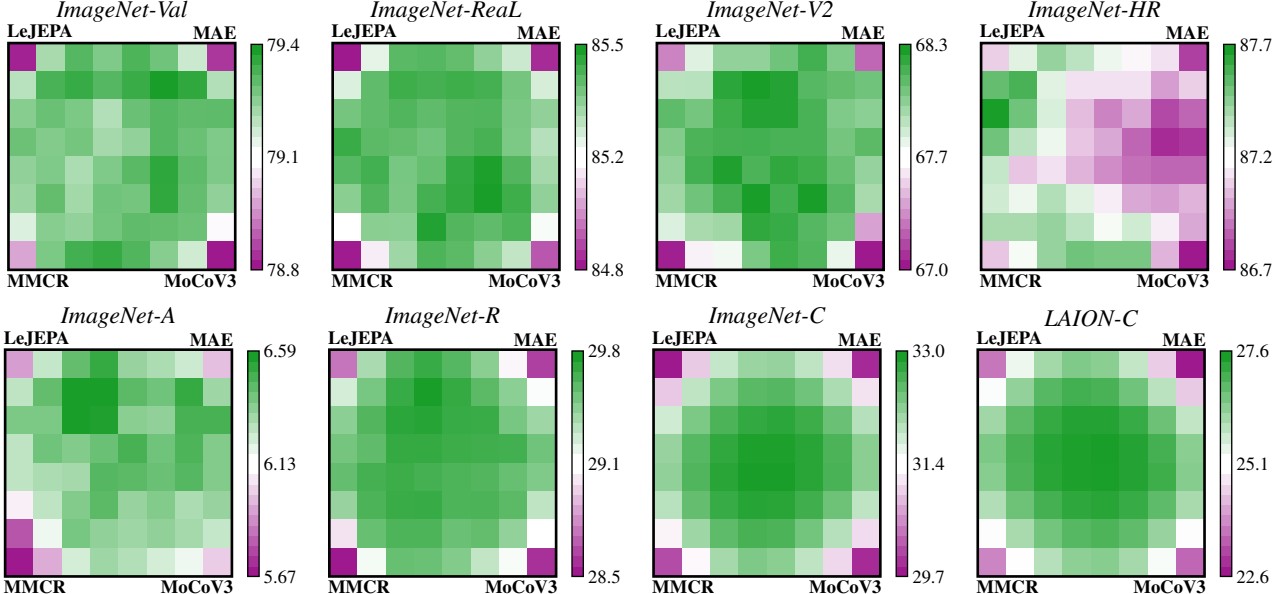

*Figure 2.* **Souping across different SSL algorithms.** We explore mixtures of 4 ingredient models that we inter-train with different self-supervised losses: LeJEPA, MAE, MoCoV3, and MMCR. MAE is a pixel-reconstruction algorithm, MoCoV3 is an instance-contrastive algorithm, and MMCR and LeJEPA are dimension-contrastive algorithms. Through this experiment, we are the first to show that linear-mode connectivity can hold between ingredients that differ in their *self-supervised* training runs. The color and shade of each inner square corresponds to the accuracy of a soup made from ingredients weighted proportionally to the proximity to the corners/ingredients.

- Inter-training on *different supervised subsets*, then fine-tuning on a target task (Aminbeidokhti et al., 2025)
- Fine-tuning with *different rewards* (Ramé et al., 2023)
- Fine-tuning with *different attacks* (Croce et al., 2023)

### 2.3. Self-Supervised Learning

**Algorithms.** Self-supervised learning (SSL) is a powerful framework because it enables learning from raw/unlabeled data itself; this allows for scaling to gigantic datasets and enables deep learning for niche applications with few annotations (Balestriero et al., 2023). We highlight three popular families of SSL algorithms, which we use. (1) SSL by reconstruction (e.g. MAE (He et al., 2022) or SimMIM (Xie et al., 2022)) masks or alters parts of samples and pre-trains models to predict the originals. (2) SSL by instance-contrastive learning (e.g. MoCoV3 (Chen et al., 2021) or SimCLR (Chen et al., 2020)) pre-trains models to produce similar embeddings derived from positive pairs of inputs (e.g. different augmentations of the same sample) and dissimilar embeddings for negatives (e.g. different samples). (3) SSL by dimension-contrastive learning (e.g. MMCR (Yerxa et al., 2023) or VICReg (Bardes et al., 2021)) pre-trains models to embed positive pairs similarly while encouraging embeddings to vary over a batch of samples without defining negative pairs. Unsurprisingly, different SSL algorithms learn different representations (Park et al., 2023) that transfer differently—motivating our use of different algorithms to create diverse ingredients that make robust soups. We

choose MAE, MoCoV3, and MMCR to represent the SSL families, and add LeJEPA (Balestriero & LeCun, 2025) as a very fresh method; we use these to train ingredient models for tasty Self-Soups. We mostly focus on MAE, since it is the most popular and accessible SSL algorithm (e.g. it is insensitive to batch size, robust to hyperparameters, computationally efficient, and applicable to many modalities).

**Algorithmic Hyperparameters.** Each SSL algorithm has its own configuration space. For example, the masking ratio in MAE, the temperature in MoCoV3, and the local-global losses in MMCR. These algorithmic choices let us train ingredient models differently to make diverse Self-Soups.

**Continued SSL** initializes from a model pre-trained by SSL on one dataset, then further trains by SSL on another dataset (Reed et al., 2022; Gururangan et al., 2020) to create domain-specific SSL models (Rodas et al., 2025; Kyrollos et al., 2023). Continued SSL is a special case of our inter-trainings varying the data, SSL algorithm, and SSL hyperparameters.

## 3. Method: Cooking without Labels and Instant Seasoning

We introduce **Self-*Soup*ervision**, which creates ingredients (parameters to mix) by training from a stock (parameters to initialize) without requiring labels at every stage. Our framework is broad; any soup that is made using ingredients that differ in their SSL runs qualifies as a Self-Soup; thus, there

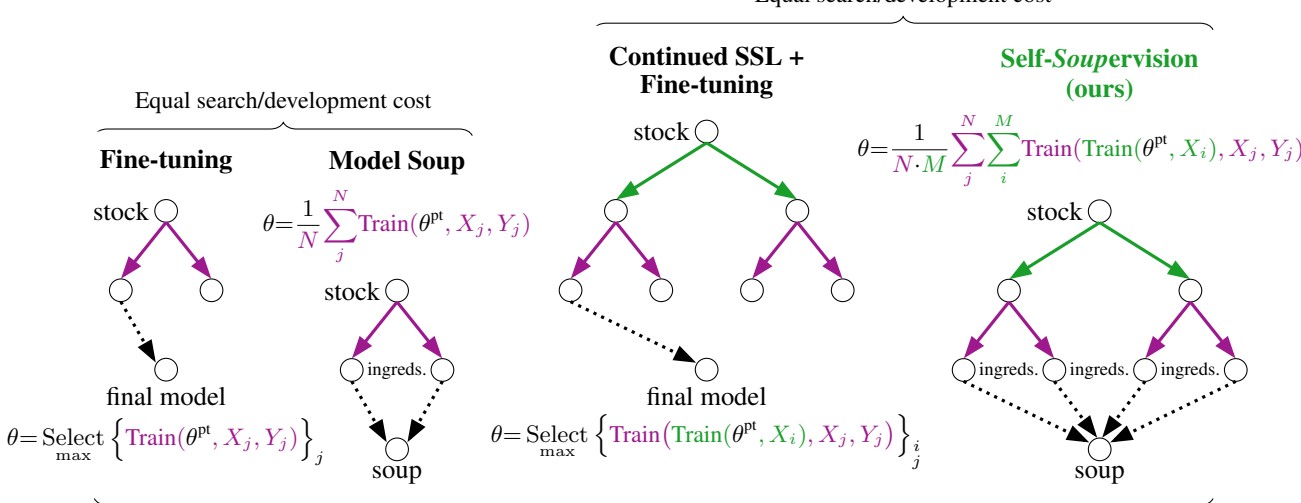

*Figure 3.* Model Soup mixes ingredients from a fine-tuning hyperparameter search; likewise, our Self-*Soup*ervision mixes ingredients from a continued SSL + fine-tuning hyperparameter search. Continued SSL enables learning from unlabeled data—e.g., from the fine-tuning domain or test-set shift—before fine-tuning. Self-Souping these diverse ingredients further improves accuracy. Symbols follow Fig. 1.

are endless possible instantiations of Self-*Soup*ervision. For example, the choice of stock, SSL algorithm and algorithmic hyperparameters, training data, training length/schedule, optimization hyperparameters, additional training stages (e.g. fine-tuning), etc. Centrally, Self-*Soup*ervision allows for cooking soups on more data and from new sources—e.g. that are closer to the target distribution—and using different losses—e.g. that are more aligned with the target task. Formally, we define Self-*Soup*ervision as:

$$\theta = \frac{1}{N \cdot M} \sum_j^N \sum_i^M \text{Train}( \overbrace{\text{Train}(\theta^{\text{pt}}, X_i)}^{Y_i \text{ not required}}, X_j, Y_j) \quad (2)$$

where $\theta$ are soup parameters, $\theta^{\text{pt}}$ are pre-trained / stock parameters, $N$ is the number of fine-tunings per inter-training, $M$ is the number of inter-trainings, $X_i / X_j$ are the inputs of the $i^{th}/j^{th}$ dataset, and $Y_j$ are the labels of the $j^{th}$ dataset. **Supervised** ingredients alone make standard soups. We introduce the **self-supervised** ingredients—which do not need labels—and which we can also use alone (i.e., by dropping the **supervised** ingredients) to make soups without fine-tuning to a task. We show these fully self-supervised soups for transfer in §4.6.

**Mixing by Self-Seasoning.** The original and simplest way to mix a soup is to average the ingredients. This uniform mixture may improve predictions, but may not be the best mixture for a given task and dataset. Seasoning (Croce et al., 2023) searches for a better mixture over a grid of options by mixing each model, making predictions on a few-shot

labeled dataset, and picking the best. While effective, this only applies to fine-tuned ingredients: seasoning makes predictions by mixing the classifiers. We instead mix purely self-supervised ingredients into a model for representation—rather than classification—then compute its representation on training and testing data for prediction by nearest neighbors. We also remove the need for seasoning to have labeled data by pairing our new ingredients with a new and unsupervised way to mix: Self-Seasoning. We optimize our $M$ mixture coefficients $\boldsymbol{\alpha} \in \mathbb{R}^M$ by gradient descent to minimize the entropy of predictions by nearest neighbors.

$$s_{ij} = \bar{\mathbf{z}}_i^\top \bar{\mathbf{z}}_j, \qquad p_{ij} = \frac{\exp(s_{ij} / T)}{\sum\limits_{j' \in \mathcal{N}_k(i)} \exp(s_{ij'} / T)}$$

$$(3)$$

$$\boldsymbol{\alpha}^* = \arg\min_{\boldsymbol{\alpha} \in \mathbb{R}^M} \frac{1}{B} \sum_{i=1}^B \left( - \sum_{j \in \mathcal{N}_k(i)} p_{ij} \log p_{ij} \right)$$

where $\bar{\mathbf{z}}_i = \mathbf{z}_i / \|\mathbf{z}_i\|_2$ is the L2-normalized embedding of sample $i$, $s_{ij}$ is the cosine similarity between samples $i$ and $j$, $\mathcal{N}_k(i)$ is the set of $k$ nearest neighbors of sample $i$ (excluding itself), $T$ is a temperature parameter, $B$ is the batch size, and $\boldsymbol{\alpha}^*$ are the mixture coefficients.

## 4. Experiments

**Baselines: Supervised soups and continued SSL.** We run five experiments to evaluate Self-Souping in different settings. First, we investigate if Self-Souping is possible

*Table 1.* **Soups on ImageNet.** Our Self-Soup gains over supervised soups on corruptions (+0.9% on IN-C and +0.9% on LAION-C). For "greedy search" and "best ingredient", we select based on IN-*Val* accuracy, so greedy may not always rank first. We include an ensemble of all ingredients run as individual models, thus is more expensive. "IN" is for ImageNet. Results are % top-1 acc. **Best** and 2ⁿᵈ best.

| Soup Type | Mix Method | IN-Val | IN-ReaL | IN-V2 | IN-HR | IN-A | IN-R | IN-C | LAION-C |
|---|---|---|---|---|---|---|---|---|---|
| Supervised Soup | Best Ingredient | 79.05 | 85.07 | 67.51 | 87.34 | 7.49 | 28.84 | 28.74 | 18.67 |
| | Greedy Search | 79.34 | 85.58 | 68.09 | 87.40 | 7.91 | 29.92 | 30.88 | 21.02 |
| | Uniform Mix | 79.12 | 85.54 | 68.01 | 87.46 | 7.75 | 30.05 | 30.91 | 22.05 |
| | *Ensemble* | *80.09* | *86.17* | *68.74* | *88.44* | *7.64* | *30.24* | *30.04* | *20.10* |
| Continued SSL + Supervised Soup | Best Ingredient | 78.98 | 85.10 | 67.59 | 87.24 | 7.16 | 29.01 | 28.56 | 18.39 |
| | Greedy Search | 79.38 | 85.66 | 68.22 | **87.60** | 8.15 | 29.70 | 31.04 | 20.66 |
| | Uniform Mix | 79.21 | 85.68 | 68.16 | 87.20 | 7.99 | 29.93 | 31.44 | 21.50 |
| | *Ensemble* | *80.05* | *86.09* | *68.92* | *88.16* | *7.76* | *30.27* | *30.38* | *19.83* |
| **Self-Soup (ours)** | Best Ingredient | 79.18 | 85.10 | 67.24 | 87.28 | 7.20 | 29.04 | 28.55 | 18.67 |
| | Greedy Search | **79.41** | **85.74** | **68.45** | 87.38 | **8.25** | 29.90 | 31.75 | 21.20 |
| | Uniform Mix | 79.09 | 85.64 | 68.07 | 87.30 | 7.83 | **30.15** | **32.34** | **22.92** |
| | *Ensemble* | *80.49* | *86.51* | *69.60* | *88.50* | *7.84* | *30.85* | *31.41* | *20.82* |

(§4.1). In the next three settings (§4.2, §4.3, §4.4), we compare against two baselines: Model Soups, which are supervised-only soups, and "continued SSL + supervised soups". The latter is a new baseline that we provide, which is a combination of existing methods yet is not a Self-Soup by our definition. A continued SSL + supervised soup first inter-trains several models using SSL, fine-tunes from them, then mixes several fine-tunings that originate from *one* inter-training; since these ingredients differ only in their *supervised* fine-tunings, they are *not* Self-Soups and are thus appropriate baselines. Continued SSL + supervised soup has an equal search cost to our Self-Soups, as they both search over $M$ SSL inter-trainings and $N$ supervised fine-tunings: please see Fig. 3. Importantly, many real-world settings prioritize a model's accuracy versus inference/deployment-cost trade-off. This prioritization is common when models are used frequently when deployed, so the cost of running them matters far more than the cost of developing them. In these settings with larger search/development budgets, our results may leave even better ingredients on the table, since Self-*Soup*ervision provides more dimensions to search and soup over (e.g. SSL algorithms and unlabeled datasets) for further gains over supervised soups. Our final setting (§4.6) directly mixes SSL trainings (without fine-tuning for a task), we compare to fine-tuning all stock parameters.

**Datasets: ImageNet and VTAB with shifts.** We choose the gold-standard for image classification, ImageNet-1K (Russakovsky et al., 2015), and the popular transfer dataset, VTAB (Zhai et al., 2020), which is a collection of 21 datasets. We evaluate extra test sets for ImageNet: ImageNet-ReaL (improved labels for ImageNet-Val (Beyer et al., 2020)), ImageNet-V2 (reproduction of ImageNet-Val (Recht et al., 2019)), ImageNet-A (challenging images (Hendrycks et al., 2021b)), ImageNet-HR (higher-effort an-

notations (Fuller et al., 2024)), ImageNet-R (rendition shifts (Hendrycks et al., 2021a)), ImageNet-C (corruption shifts (Hendrycks & Dietterich, 2019)), and LAION-C (more corruption shifts (Li et al., 2025)). VTAB does not have shifts, so we make our own shifted data with ImageNet-C's code for all 15 corruptions types at the highest severity. We make 1K subsets of the train and test sets for convenient computation. We call our version *mini*-VTAB-*C*, as it is smaller and has corruptions, and it is shared in the supplement (§D).

**Stocks.** We use the MAE stock (ViT-B pre-trained for 1600 epochs on ImageNet-1K by He et al. (2022)). Later (Fig. 5) we show Self-Souping works equally well for another stock.

### 4.1. Is Self-*Soup*ervision possible and productive?

**Setup: Souping over different SSL algorithms.** To check if ingredients can be souped that differ in their self-supervision, we first inter-train 4 models on the ImageNet-1K training set for 5 epochs with a 1e-5 learning rate. One model uses the MAE algorithm, another uses MoCoV3, another MMCR, and the last uses LeJEPA. After inter-training, we fine-tune each model using supervised learning for 10 epochs (following Wortsman et al. (2022)) with an 8e-5 learning rate. After fine-tuning, we have 4 ingredients from which to cook our soup. To explore the mixture space, we compute 64 convex combinations of ingredients that are uniformly distributed. For each of the 64 models, we test on the ImageNet test sets.

**Results: Self-Souping is productive and LMC holds.** Fig. 2 shows 64 different mixtures of our 4-ingredient soups across 8 ImageNet test sets. For all test sets, souping across SSL algorithms is productive: the **best** models are always a combination of ingredients and the **worst** models are always the ingredients alone. The largest gains of +3% are on

corrupted data (ImageNet-C and LAION-C), and the best mixtures are roughly-equal combinations of our 4 ingredients (i.e. the central squares). LMC can thus hold between self-supervised ingredients—a novel finding that whets the appetite for more tasty soups now that it is possible.

### 4.2. Can Self-Soups outperform supervised soups?

**Setup: Inter-train on ImageNet then fine-tune on ImageNet.** We experiment with SSL inter-training on the fine-tuning data. In this case, self-supervision provides more ingredients, and more diverse ingredients by inter-training using different SSL algorithms. We fine-tune each of the 4 inter-trainings from §4.1 for 10 epochs—doing this 4 times (varying fine-tuning learning rates {6e-5, 8e-5, 1e-4, 1.5e-4}). We compare to initialization from the MAE stock.

**Results: Self-Soups help on corruptions.** Self-Souping's largest gains are on corrupted data: +0.9% on ImageNet-C and +0.9% on LAION-C over other soups (Tab. 7). Our Self-Soups are more robust than an ensemble, which averages logits across ingredients, by 0.9% on ImageNet-C and 2.1% on LAION-C.

### 4.3. Does inter-training on the test distribution help?

*Table 2.* **Self-*Soup*ervision on the test distribution provides large gains: +3.5% on IN-C and +7% on LAION-C.** These soups mix ingredients inter-trained by SSL on *unlabeled* test data. In the disjoint setting, we use even-indexed test samples for inter-training on corruptions and odd-indexed samples for testing on corruptions. In the joint setting, we use even-indexed test samples for both inter-training and testing. "IN" is for ImageNet, "LA" is for LAION. Results are % top-1 acc. **Best** result and 2[nd] best.

| Method | IN-Val | IN-C | LA-C |
|---|---|---|---|
| *Same models as Tab. 7 for reference* | | | |
| Supervised Soup on IN-Train | 79.12 | 30.91 | 22.05 |
| Cont. SSL + Soup on IN-Train | 79.21 | 31.44 | 21.50 |
| Self-Soup on IN-Train | 79.09 | 32.34 | 22.92 |
| *Disjoint (inter-train: even-indexed, test: odd-indexed samples)* | | | |
| Cont. SSL + Soup on IN-C | **79.16** | 35.09 | 22.63 |
| Self-Soup on IN-C | 78.96 | **35.72** | 23.28 |
| ↪ Best ingredient | 78.91 | 31.41 | 19.55 |
| Cont. SSL + Soup on LA-C | 79.15 | 31.65 | 27.43 |
| Self-Soup on LA-C | 78.87 | 32.58 | **29.48** |
| ↪ Best ingredient | 78.87 | 28.25 | 23.38 |
| Self-Soup on IN-C + LA-C | 78.81 | 34.43 | 26.39 |
| *Joint (inter-train: even-indexed, test: even-indexed samples)* | | | |
| Self-Soup on IN-C | **78.96** | **36.02** | 23.51 |
| ↪ Best ingredient | 78.91 | 31.70 | 19.57 |
| Self-Soup on LA-C | 78.87 | 32.51 | **30.32** |
| ↪ Best ingredient | 78.87 | 28.09 | 23.84 |
| Self-Soup on IN-C + LA-C | 78.81 | 34.55 | 26.83 |

**Setup: Inter-train on shifts then fine-tune on ImageNet.** We now allow for SSL inter-training on shifted data. In this case, SSL enables ingredients to learn from the test distribution *without labels* for robustness to it. We measure this on split data for optimization and evaluation (Tab. 2). We first inter-train 4 models (varying learning rates {1e-5, 2e-5, 3e-5, 4e-5}) by MAE for 100K steps on unlabeled test samples that are *even-indexed*. We fine-tune these models back on the ImageNet training set following the fine-tuning runs in §4.2. We report results for *odd-indexed* and *even-indexed* test samples to measure accuracy when inter-training and evaluation samples are disjoint and joint, respectively.

**Results: Shift-aware ingredients deliver robustness.** Self-Souping on the test distribution (but not the test samples themselves) provides large gains: +3.5% on ImageNet-C and +7% on LAION-C (Tab. 2). Self-Souping on test samples themselves provides a small boost on top: +0.3% on ImageNet-C and +0.8% on LAION-C. Despite the different types of corruptions present in ImageNet-C versus LAION-C, there are benefits to inter-training on one set of corruptions to the other set. Self-Souping over both test sets—i.e. where ingredients differ in their SSL inter-training data distributions and fine-tuning runs—keeps most of the shift-specific gains.

*Table 3.* **Our Self-Soup cooked on ImageNet-C keeps its robustness advantage after test-time adaptation (TTA).** We run soups prepared on ImageNet-Train (from Tab. 7 and 2) on ImageNet-C (odd indices) with and without TTA—and repeat for our Self-Soup on ImageNet-C (*disjoint*, from Tab. 2). Results are % top-1 acc.

| Method | without TTA | with TTA |
|---|---|---|
| Supervised Soup on ImageNet-Train | 30.95 | 32.67 |
| Self-Soup on ImageNet-Train | 32.38 | 34.13 |
| Self-Soup on ImageNet-C | 35.72 | 37.50 |

**Bonus: Why not adapt to the shift at test time?** Another unsupervised way to adapt to a shift is test-time adaptation (TTA), e.g. by minimizing prediction entropy (Wang* et al., 2021). We use SAR (Niu et al., 2023) as a SOTA TTA method. Our Self-Soup on ImageNet-C without SAR still beats soups prepared on ImageNet-Train with SAR applied on ImageNet-C (Tab. 3). In this case, our soup made from shift-aware ingredients that we updated on the test distribution (odd-indexed), not test samples (even-indexed), outperforms models updated on test samples. This Self-Soup on ImageNet-C gains more with adaptation by SAR ($35.72 \rightarrow 37.50$), showing the two strategies can be complementary.

### 4.4. Can Self-*Soup*ervision bring transfer gains?

**Setup: Self-Souping on 21 downstream tasks.** We cook task-specific Self-Soups for improved in- and out-of-

*Table 4.* **Soups for transfer.** Measured across 336 test sets (21 datasets × 16 corruptions), our Self-Soups show modest yet useful gains. We make soups that are specialized for 21 tasks (from the VTAB collection) from 3 broad domains. We evaluate on clean / original test data and 15 corruption types to measure robustness. For each soup type, we evaluate the best ingredient (selected on clean test data), a greedy search over ingredients (selected on clean test data), and a uniform mixture of all ingredients. We emphasize the **best** and 2nd best.

*(a)* Top-1 % accuracy for **21 tasks** (VTAB) averaged over 15 corruption types and clean data. Self-Soups are most helpful on natural data.

| | | Natural | | | | | | | | Specialized | | | | Structured | | | | | | | | | |
| --- | --- | --- | --- | --- | --- | --- | --- | --- | --- | --- | --- | --- | --- | --- | --- | --- | --- | --- | --- | --- | --- | --- | --- |
| Soup Type | Mix Method | Caltech101 | CIFAR-10 | CIFAR-100 | DTD | Flowers102 | Pets | Sun397 | SVHN | Camelyon | EuroSAT | Resisc45 | Retinopathy | Clevr-Count | Clevr-Dist | DMLab | dSpr-Loc-X | dSpr-Loc-Y | dSpr-Loc-Ori | KITTI-Dist | sNORB-Azim | sNORB-Elev | Mean |
| Supervised Soup | Best ingredient | 39.4 | 58.4 | 28.1 | 24.2 | 26.9 | 34.9 | 7.4 | 57.5 | 67.2 | 49.8 | 29.7 | 67.3 | 30.0 | 26.1 | 35.1 | 8.1 | 16.0 | 24.1 | 48.0 | 9.4 | 18.3 | 33.6 |
| | Greedy Search | 41.1 | 58.6 | 29.1 | 26.1 | 28.2 | 35.6 | 8.3 | 57.8 | 67.2 | 49.8 | 30.8 | 68.1 | 30.0 | 26.1 | 35.1 | 8.2 | 15.5 | 23.4 | 52.6 | 9.5 | 19.5 | 34.3 |
| | Uniform Mix | 42.5 | 58.1 | 29.6 | 26.0 | 27.2 | 35.0 | 8.2 | 56.6 | 66.4 | 50.7 | 30.8 | 69.3 | 31.5 | 29.7 | 36.4 | 7.7 | 16.2 | 23.0 | 53.4 | 9.2 | 19.3 | 34.6 |
| Continued SSL + Supervised Soup | Best ingredient | 40.8 | 56.8 | 27.7 | 24.2 | 27.4 | 31.8 | 8.6 | 55.5 | 65.8 | 49.8 | 27.7 | 71.2 | 29.3 | 24.0 | 33.6 | 6.3 | 13.9 | 22.3 | 47.5 | 10.2 | 17.8 | 33.0 |
| | Greedy Search | 40.8 | 58.4 | 29.1 | 24.2 | 28.3 | 34.1 | 8.9 | 57.5 | 66.4 | 49.8 | 29.8 | 71.2 | 29.3 | 24.0 | 34.8 | 6.3 | 14.4 | 23.6 | 50.9 | 10.2 | 17.8 | 33.8 |
| | Uniform Mix | 44.1 | 58.3 | 29.5 | 24.9 | 27.9 | 34.2 | 8.9 | 58.3 | 67.3 | 49.8 | 29.8 | 71.3 | 28.4 | 28.7 | 35.5 | 6.5 | 14.2 | 23.1 | 49.8 | 10.1 | 19.4 | 34.3 |
| **Self-Soup (ours)** | Best ingredient | 40.3 | 56.8 | 27.7 | 24.2 | 28.1 | 34.1 | 8.6 | 59.9 | 66.4 | 49.8 | 30.0 | 68.1 | 28.6 | 24.0 | 33.6 | 6.6 | 13.0 | 24.8 | 43.7 | 10.0 | 18.5 | 33.3 |
| | Greedy Search | 42.1 | 59.8 | 30.8 | 25.2 | 29.6 | 36.4 | 9.2 | 59.9 | 67.1 | 49.8 | 29.8 | 68.1 | 28.7 | 26.2 | 34.8 | 6.6 | 13.8 | 26.0 | 49.5 | 10.0 | 18.6 | 34.4 |
| | Uniform Mix | 44.6 | 59.7 | 30.6 | 26.1 | 29.5 | 36.4 | 8.9 | 58.4 | 66.9 | 50.0 | 31.3 | 70.2 | 29.8 | 29.4 | 36.2 | 6.9 | 14.6 | 25.6 | 52.1 | 10.2 | 19.9 | 35.1 |

*(b)* Top-1 % accuracy for **15 corruption types and clean data** averaged over 21 tasks (VTAB). Self-Soups help most against blur, weather, and digital corruptions. The plain supervised soup (Wortsman et al., 2022) is closest to the MAE stock and is best against noise.

| | | Clean | Noise | | | Blur | | | | Weather | | | | Digital | | | | Mean |
| --- | --- | --- | --- | --- | --- | --- | --- | --- | --- | --- | --- | --- | --- | --- | --- | --- | --- | --- |
| Soup Type | Mix Method | | Gaussian | Shot | Impulse | Defocus | Glass | Motion | Zoom | Snow | Frost | Fog | Brightness | Contrast | Elastic | Pixel | JPEG | |
| Supervised Soup | Best ingredient | 64.3 | 15.6 | 17.5 | 15.1 | 37.5 | 34.3 | 35.8 | 42.2 | 30.5 | 31.1 | 29.8 | 55.1 | 18.6 | 38.4 | 35.1 | 37.1 | 33.6 |
| | Greedy Search | 64.7 | 16.5 | 18.3 | 16.1 | 39.1 | 34.0 | 36.8 | 42.4 | 31.0 | 31.1 | 30.2 | 55.9 | 19.9 | 39.5 | 35.4 | 38.1 | 34.3 |
| | Uniform Mix | 63.9 | 16.1 | 18.3 | 16.2 | 40.2 | 35.4 | 38.1 | 42.2 | 30.4 | 31.0 | 30.6 | 55.8 | 21.3 | 39.9 | 36.3 | 38.1 | 34.6 |
| Continued SSL + Supervised Soup | Best ingredient | 65.7 | 14.0 | 16.6 | 13.9 | 36.7 | 32.7 | 37.6 | 42.3 | 28.6 | 29.0 | 29.3 | 53.3 | 19.6 | 38.5 | 34.1 | 35.3 | 33.0 |
| | Greedy Search | 66.2 | 15.1 | 17.4 | 14.8 | 37.8 | 33.9 | 38.2 | 42.5 | 29.7 | 30.2 | 30.0 | 54.1 | 20.2 | 39.7 | 34.7 | 36.2 | 33.8 |
| | Uniform Mix | 65.2 | 14.8 | 17.7 | 14.7 | 39.1 | 35.0 | 38.7 | 42.8 | 30.1 | 30.7 | 30.9 | 55.5 | 21.0 | 41.1 | 35.2 | 36.1 | 34.3 |
| **Self-Soup (ours)** | Best ingredient | 66.4 | 14.0 | 16.3 | 13.8 | 37.1 | 30.8 | 36.7 | 42.9 | 30.4 | 29.4 | 29.0 | 55.0 | 20.8 | 38.8 | 34.0 | 35.4 | 33.2 |
| | Greedy Search | 67.5 | 14.3 | 17.3 | 14.1 | 38.8 | 32.5 | 38.8 | 43.7 | 30.5 | 31.2 | 31.1 | 56.7 | 21.1 | 40.6 | 34.7 | 37.1 | 34.4 |
| | Uniform Mix | 65.6 | 14.7 | 18.0 | 14.9 | 40.3 | 35.4 | 39.9 | 43.8 | 31.2 | 31.8 | 31.7 | 56.3 | 22.6 | 41.6 | 36.4 | 37.4 | 35.1 |

distribution accuracy. In this case, SSL allows ingredients to learn from unlabeled samples from the target distribution prior to fine-tuning. For each of the 21 datasets in VTAB, we inter-train 4 models (varying learning rates {1e-5, 2e-5, 3e-5, 4e-5}) using MAE for 10K steps on *all* training samples. For each model, we fine-tune 4 times (varying learning rates {1e-5, 2e-5, 3e-5, 4e-5}) for 100 epochs on mini-VTAB. As a baseline, we run the same fine-tuning procedure but initialize from the pre-trained MAE stock.

**Results: Self-Souping transfers well.** Averaged over the 21 mini-VTAB datasets, our Self-Soup with greedy search achieves 67.5% top-1 accuracy on clean test data, the second best achieves 66.2% (Tab. 4). If we then average over the corrupted test sets in mini-VTAB-C, our Self-Soup with uniform mixing achieves 35.1%, the next best achieves 34.6%. Self-Souping thus provides small yet useful gains that we thoroughly measure across 336 test sets (21 tasks × 16 corrupt/natural conditions). Our method gains the most for the 8 natural datasets: +1% over the continued SSL + su-

pervised soup, and +1.4% over the supervised-only soup. Self-Souping is most robust to blur, weather, and digital corruption types (+2%), while it handles noise worse (-1% versus the supervised-only soup).

## 4.5. Preparing ingredients *with* versus *without* labels

**Setup: Self-Soup versus Model Ratatouille.** We now check if Self-Souping can compete with Model Ratatouille, which inter-trains with supervision on auxiliary labeled datasets before fine-tuning on the target task and mixing ingredients to make a soup. We inter-train 4 times using the natural tasks from VTAB (Flowers (Nilsback & Zisserman, 2008), Pets (Parkhi et al., 2012), Caltech101 (Fei-Fei et al., 2006), and Sun397 (Xiao et al., 2010)) by supervised learning. Then for each inter-trained model, we fine-tune on ImageNet-1K 4 times using learning rates {6e-8, 8e-8, 1e-4, 1.5e-4} and uniformly mix all 16 ingredients. We compare this supervised soup with our Self-Soup that we first inter-train 4 times using SSL algorithms (MAE, Mo-

CoV3, MMCR, LeJEPA) on ImageNet-1K, then fine-tune each model 4 times using the same learning rates and uniformly mix all 16 ingredients. We test on ImageNet-Val, ImageNet-C, and LAION-C.

*Table 5.* **Our Self-Soup matches Model Ratatouille (Ramé et al., 2023)** *without needing auxiliary labeled datasets.* We prepare Ratatouille ingredients on 4 different labeled datasets before fine-tuning on ImageNet-1K then mixing uniformly. We prepare Self-Soup ingredients by varing SSL algorithms on ImageNet-1K before fine-tuning and mixing. Results are % top-1 acc.

| Method | Needs extra labels | IN-Val | IN-C | LAION-C |
|---|---|---|---|---|
| Ratatouille | ✔ | 79.05 | 32.20 | 22.89 |
| Self-Soup | ✘ | 79.09 | 32.34 | 22.92 |

**Results: Self-Souping matches Ratatouille.** Both approaches are equally accurate (Tab. 5), so our Self-Soup is a practical improvement because it is more general. Self-Souping is more general since it (1) does not need these extra labeled datasets, (2) can harness unlabeled data, and (3) mix without task-fine-tuning for fully unsupervised soups.

### 4.6. Can we mix ingredients directly after SSL inter-training for few-shot classification?

**Setup: Self-Seasoning of self-supervised ingredients.** We now mix soups without supervised fine-tuning and evaluate by nearest neighbors (kNN). We make 6 ingredients per task with MAE, MMCR ($\times 2$), MoCoV3 ($\times 2$), and the stock. For the 5 task-specific ingredients, we train for 10K steps with a 4e-5 learning rate on the full training data for each VTAB task. To find mixture coefficients we Self-Season our Self-Soup by minimizing kNN entropy on the mini-VTAB training data *without* labels. To put our Self-Seasoning results in context, run two baselines. We call the first baseline "labeled seasoning" since it minimizes the supervised cross-entropy loss of kNN prediction. kNN prediction is needed since these are purely self-supervised ingredients without task-specific classifying heads. Labeled seasoning makes predictions in the same way as Self-Seasoning, learns the same parameters (the 6 mixture coefficients), and uses the same ingredients—our experiment thus controls for these factors. Our second baseline simply fully fine-tunes the stock. In all 3 settings, we train for 100 epochs and search through 6 hyperparameter settings. Please see the supplement for more details (§ E).

**Results: Seasoning can outperform fine-tuning.** For natural tasks, both seasoning methods outperform fine-tuning when the number of training samples is <500 (Tab. 4). Both of the resulting seasoned soups are Self-Soups because they are made from SSL ingredients. This shows another use-case for our Self-Soups, i.e. when labeled data is limited. Our experiments also show that not only can we make tasty

ingredients without labels, but we can also season to make tasty soups without labels: Self-Seasoning is competitive with labeled seasoning.

### 4.7. What about other stocks?

There is no reason for our Self-Soups to require an MAE stock. To show another, we inter-train separately using MAE and MoCoV3 on ImageNet—initializing from Franca's pre-trained ViT-B (a strong open-source model (Venkataramanan et al., 2025)). After inter-training, we fine-tune on ImageNet, then mix, showing that LMC holds and gains are similar to an MAE stock (Fig. 5).

## 5. Related Work

**Sophisticated Mixing or Merging.** Model stock (Jang et al., 2024) refines the mixing of soups with layer-wise re-weighting using the angles between ingredient parameters. Complementarily, our Self-Souping provides more *ingredients* that are compatible with more sophisticated mixing. Model *merging* methods (e.g. TIES-Merging (Yadav et al., 2023) or EMR-Merging (Huang et al., 2024)) combine multiple models, like soups. These methods merge models for different tasks (e.g. text summarization and translation), which do not share an initialization, and may not even share a common architecture. Merging methods do not necessarily maintain the computational cost of their input models, in contrast to soups in general and our Self-Soups in particular.

**Multi-Task SSL.** Our inter-trainings each optimize their own model with a single self-supervised loss. Multi-task SSL instead jointly optimizes a shared model with multiple self-supervised losses (Doersch & Zisserman, 2017; Bachmann et al., 2022). Although such multi-task optimization can improve on single task optimization, it can require larger-scale computational resources to achieve sufficient batch sizes (at least multiple GPUs, if not multiple machines) and more tuning to balance losses and gradients. While Self-Soups require multiple inter-trainings, each experiment is simpler and smaller-scale. We could do both and mix multi-task SSL ingredients as a Self-Soup by definition.

**Domain Adaptation.** Our inter-training on shifts is related to unsupervised domain adaptation (UDA): joint optimization on labeled "source" data and unlabeled "target" data (Saenko et al., 2010). However, our inter-trainings are simpler independent training runs, rather than joint optimizations, and are computationally more efficient in only updating on the target data. Test-time adaptation and test-time training (TTA/TTT) make predictions and update on the target data at the same time by online optimization. These updates can alter statistics (Schneider et al., 2020) and model parameters without supervision (Sun et al., 2020; Wang* et al., 2021). While such test-time updates can be

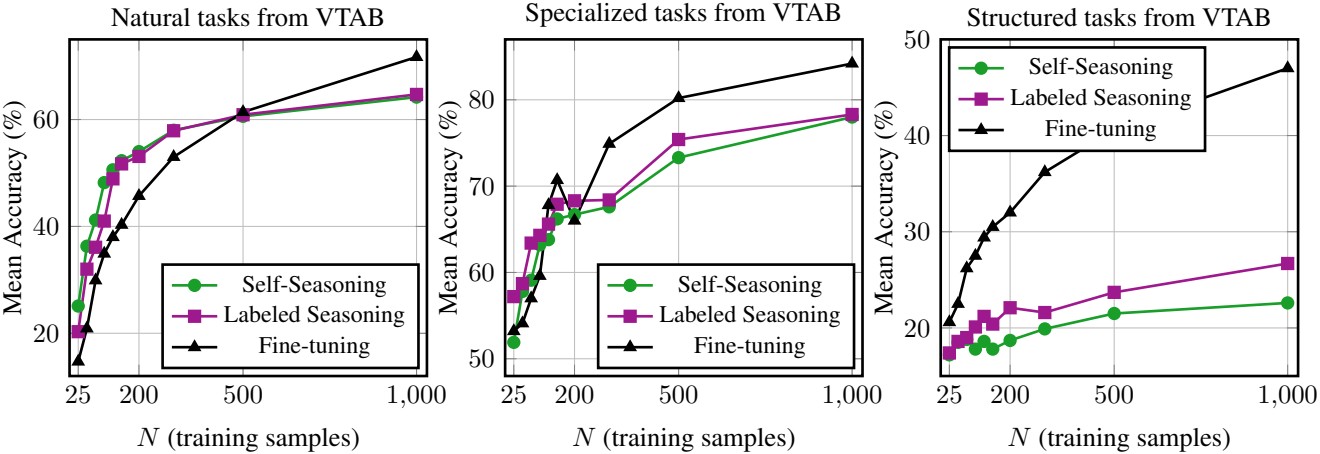

*Figure 4.* Our Self-Seasoning is competitive with labeled seasoning on VTAB datasets. Self-Seasoning learns soup-mixture coefficients without labels by minimizing the kNN entropy of each batch during training. These coefficients weight our SSL-only ingredients in the soup. Both seasoning methods, which only learn 6 parameters, outperform fine-tuning with less than 500 training samples on natural tasks (left). For each dataset, we train for 100 epochs and give each method an equal hyperparameter-tuning budget.

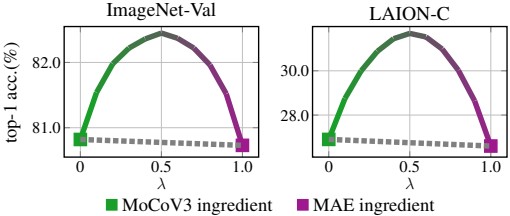

*Figure 5.* **Self-Souping is possible and productive for another stock: Franca.** (Venkataramanan et al., 2025) Mixing models independently trained using *SSL* (= ingredients) improves accuracy over ingredients alone with the Franca ViT-B stock.

efficient and effective, they need careful tuning and more test-time computation. After inter-training and mixing, Self-Soups are deployed without more test-time computation.

## 6. Discussion

**Limitations.** Self-Soups enlarge the soup kitchen (SSL methods, hyperparameters, and data) with our new recipes, but there are more to cook. Our largest gains (+7%) need unlabeled target/shifted data, which may not be available. Other gains are modest ($\lesssim 1\%$) yet useful, as they do not raise inference costs. There are dozens of SSL algorithms absent from our study, but we choose from 3 different SSL families, so our findings may generalize within families.

**Conclusion.** We introduce Self-*Soup*ervision, which generalizes model soups to SSL. Self-Souping adds to the menu by harnessing different losses to flavor ingredients from different distributions without requiring labels. We first show that mixing ingredients that differ in their self-supervised training runs (e.g. different losses) is possible and produc-

tive. We then show that Self-Soups can improve supervised soups on ImageNet and VTAB. Self-Souping is most helpful when facing distribution shifts—and especially, when unlabeled shifted data is available for preparing ingredients. We also introduce Self-Seasoning, which learns ingredient mixtures for a task without training labels. We hope our recipes earn a spot in your cookbook and inspire new ones.

## Acknowledgements

We thank Simon Ghyselincks, Pritam Sarkar, and Pierre Lardet for pre-reviewing the manuscript. AF is primarily supported by an NSERC PGS-D scholarship. ES is supported by a Canada CIFAR AI Chair. Resources used in preparing this research were provided, in part, by the Province of Ontario, the Government of Canada through CIFAR, and companies sponsoring the Vector Institute.

## Impact Statement

Our Self-*Soup*ervision method and ingredients aim to produce more machine learning models and more accurate models. Improving the generalization and robustness of machine learning models contributes to their accuracy and sound deployment in practice. We evaluate on standard benchmarks for visual recognition, and so do not alter the choice of tasks for better or worse. Our use of self-supervised learning without labels is potentially more general and feasible for a broader set of applications, because self-supervised ingredients can be inter-trained without the cost of annotation, though our soups do still require the cost of computation. The workflow of inter-training, fine-tuning, and mixing is potentially more accessible and collaborative, because con-

tributing an inter-training or fine-tuning and evaluating a mixture are less computationally intensive than pre-training. We intend for Self-*Soup*ervision to enable more of the community to engage in machine learning.

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

# A. More ImageNet Experments.

*Table 6.* **Soups on ImageNet.** We use the same experiment setup as Tab. 7, except that we use RandAug (2 transformations with strength 15) instead of 3-Augment. Our Self-Soups remain better than alternatives.

| Soup Type | Mix Method | IN-Val | IN-ReaL | IN-V2 | IN-HR | IN-A | IN-R | IN-C | LAION-C |
|---|---|---|---|---|---|---|---|---|---|
| Supervised Soup | Best Ingredient | 78.99 | 84.92 | 67.38 | 86.84 | 6.35 | 28.17 | 29.59 | 23.16 |
| | Greedy Search | 79.22 | 85.30 | 67.89 | 87.30 | 6.29 | 28.81 | 31.79 | 26.55 |
| | Uniform Mix | 79.03 | 85.29 | 67.93 | 87.12 | 6.15 | 29.32 | 32.18 | 27.55 |
| | Ensemble | 80.11 | 85.92 | 68.86 | 88.04 | 6.23 | 29.54 | 31.96 | 24.68 |
| Continued SSL + Supervised Soup | Best Ingredient | 78.97 | 84.80 | 67.06 | 87.08 | 5.67 | 28.46 | 30.10 | 23.76 |
| | Greedy Search | 79.34 | 85.34 | 67.73 | 87.28 | 6.25 | 29.46 | 32.09 | 25.96 |
| | Uniform Mix | 79.22 | 85.33 | 67.74 | 87.18 | 6.25 | 29.47 | 32.29 | 26.94 |
| | Ensemble | 80.03 | 85.79 | 68.63 | 87.96 | 6.19 | 29.56 | 32.04 | 24.71 |
| **Self-Soup (ours)** | Best Ingredient | 78.97 | 84.80 | 67.06 | 87.08 | 5.67 | 28.46 | 30.10 | 23.76 |
| | Greedy Search | 79.44 | 85.44 | 67.95 | 87.22 | 6.57 | 29.44 | 32.79 | 27.11 |
| | Uniform Mix | 79.00 | 85.35 | 67.92 | 87.06 | 6.48 | 29.82 | 33.04 | 28.22 |
| | Ensemble | 80.50 | 86.27 | 69.36 | 88.58 | 6.39 | 30.02 | 33.12 | 25.57 |

*Table 7.* **Soups on ImageNet.** We use the same experiment setup as Tab. 7, except that we make twice as many ingredients for all methods by using 3-Augment for half the ingredients and RandAug (2 transformations with strength 15) for the other half. Our Self-Soups remain better than alternatives.

| Soup Type | Mix Method | IN-Val | IN-ReaL | IN-V2 | IN-HR | IN-A | IN-R | IN-C | LAION-C |
|---|---|---|---|---|---|---|---|---|---|
| Supervised Soup | Best Ingredient | 79.05 | 85.07 | 67.51 | 87.34 | 7.49 | 28.84 | 28.74 | 18.67 |
| | Greedy Search | 79.44 | 85.62 | 68.45 | 87.70 | 7.68 | 29.62 | 32.48 | 22.86 |
| | Uniform Mix | 79.14 | 85.63 | 68.26 | 87.42 | 7.31 | 29.68 | 32.95 | 24.69 |
| | Ensemble | 80.51 | 86.38 | 69.59 | 88.68 | 7.44 | 30.62 | 32.53 | 23.49 |
| Continued SSL + Supervised Soup | Best Ingredient | 78.97 | 85.10 | 67.17 | 87.66 | 7.36 | 28.77 | 28.94 | 19.85 |
| | Greedy Search | 79.49 | 85.65 | 68.49 | 87.66 | 7.21 | 29.22 | 33.03 | 24.64 |
| | Uniform Mix | 79.30 | 85.50 | 68.20 | 87.52 | 7.12 | 29.54 | 33.08 | 24.80 |
| | Ensemble | 80.53 | 86.37 | 69.27 | 88.28 | 7.21 | 30.66 | 32.70 | 23.53 |
| **Self-Soup (ours)** | Best Ingredient | 79.18 | 85.10 | 67.24 | 87.28 | 7.20 | 29.04 | 28.55 | 18.67 |
| | Greedy Search | 79.61 | 85.85 | 68.35 | 87.36 | 7.75 | 29.60 | 33.62 | 23.90 |
| | Uniform Mix | 79.13 | 85.62 | 68.23 | 87.52 | 7.40 | 29.89 | 33.68 | 25.22 |
| | Ensemble | 80.69 | 86.54 | 69.75 | 88.58 | 7.28 | 30.87 | 33.40 | 24.44 |

# B. VTAB References.

For clarity and credit, we reference the original datasets that went into the VTAB collection: Caltech101 (Fei-Fei et al., 2006), CIFAR-10/100 (Krizhevsky et al., 2009), DTD (Cimpoi et al., 2014), Flowers102 (Nilsback & Zisserman, 2008), Pets (Parkhi et al., 2012), Sun397 (Xiao et al., 2010), SVHN (Netzer et al., 2011), EuroSAT (Helber et al., 2019), Resisc45 (Cheng et al., 2017), Patch Camelyon (Veeling et al., 2018), Retinopathy (Kaggle & EyePacs, 2015), Clevr (Johnson et al., 2017), dSprites (Matthey et al., 2017), SmallNORB (LeCun et al., 2004), DMLab (Beattie et al., 2016), and KITTI (Geiger et al., 2013).

# C. More training details.

**All runs.** We always use: 0.01 weight decay, the AdamW optimizer (Loshchilov & Hutter, 2019), warmup for 10% of the steps and cooldown via cosine decay, and 3-Augment (Touvron et al., 2022) for data augmentation.

**All SSL inter-training runs.** For MAE inter-training, we initialize the decoder with the pre-trained MAE decoder. For MoCoV3 and MMCR inter-training, we use a simple 2-layer MLP as the projection head, we do not use an exponential

moving average to compute target embeddings, and we warmup the head for only 1% of the steps (chosen so the head learns more quickly than the backbone). We choose these settings to keep it simple and do not tune them. Before mixing or fine-tuning the ingredients, we discard all algorithm-specific heads and only use the backbones/encoders.

**All ImageNet fine-tuning runs.** We fine-tune for 10 epochs on ImageNet-1K (following the original Model Soups (Wortsman et al., 2022)). We sweep learning rates {6e-5, 8e-5, 1e-4, 1.5e-4} with a 128 batch size. We always use LPFT, which initializes fine-tuning from the linear probed solution (Kumar et al., 2022) (including when fine-tuning on VTAB).

**ImageNet: §4.2.** For model inter-training, we train for 5 epochs on ImageNet-1K with a 256 batch size and a 1e-5 learning rate. For MAE, we use a 90% masking ratio and 1 decoder layer. For MoCoV3, we use a 1.0 temperature. For MMCR, we use both global and local losses. For LeJEPA, we use $\lambda$=0.02. These SSL-algorithm hyperparameters were mostly chosen arbitrarily, in our experience different choices achieves the same results.

**Test-set inter-training: §4.3.** For model inter-training, we train for 100K steps with a 128 batch size using the default MAE settings (i.e. 75% masking ratio and 8 decoder layers), and sweep learning rates {1e-5, 2e-5, 3e-5, 4e-5}.

**Test-time adaptation: Tab. 3.** We sweep base learning rates {1e-5, 3e-5, 5e-5, 8e-5, 1e-4, 3e-4, 1e-3, 3e-3}. A 5e-5 base learning rate is best. We use a 128 batch size, which sets the actual learning rate: $lr = (\text{base\_lr}/64) \cdot \text{batch\_size}$

# D. Data and Code.

**Data.**
https://huggingface.co/datasets/antofuller/mini-VTAB
https://huggingface.co/datasets/antofuller/mini-VTAB-corruptions

**Code.**
https://github.com/antofuller/self_soupervision

# E. Few-shot experiments: Self-Seasoning, Labeled Seasoning, and Fine-tuning §4.6.

For all three methods we train for 100 epochs with AdamW and a 256 batch size (or the number of training samples if $N<256$). For Self-Seasoning we run 6 times varying k {8, 16, 32} and T {0.07, 0.2} with a max learning rate of 0.1 and cosine-decay it to 0.01. For labeled seasoning we run 6 times varying the max learning rate {3e-1, 1e-1, 3e-2, 1e-2, 3e-3, 1e-3}. For fine-tuning we run 6 times varying the max learning rate {3e-3, 1e-3, 3e-4, 1e-4, 3e-5, 1e-5}. We pick the best of the 6 runs for each method on the held-out set.

The following is Self-Seasoning PyTorch code:

```
1  def knn_inbatch_neighbor_entropy(Z, k=16, T=0.07):
2      B = Z.shape[0]
3
4      # L2 normalize embeddings
5      Z = F.normalize(Z, dim=1, p=2)          # [B, D]
6
7      # Compute pairwise cosine similarities
8      S = Z @ Z.T                              # [B, B]
9      S.fill_diagonal_(float("-inf"))         # mask self
10
11     # Select top-k neighbors
12     S_topk, _ = S.topk(k, dim=1)            # [B, k]
13
14     # Compute neighbor distribution
15     p = (S_topk / T).softmax(dim=1)         # [B, k]
16
17     # Compute entropy
18     H = -(p * p.clamp(min=1e-12).log()).sum(dim=1)
19
20     return H.mean()
```

*Table 8.* Learned Self-Seasoning coefficients for each mini-VTAB dataset.

| Ingredient | Caltech101 | CIFAR-10 | CIFAR-100 | DTD | Flowers102 | Pets | Sun397 | SVHN | Camelyon | EuroSAT | Resisc45 | Retinopathy | Clevr-Count | Clevr-Dist | DMLab | dSpr-Loc-X | dSpr-Loc-Y | dSpr-Loc-Ori | KITTI-Dist | sNORB-Azim | sNORB-Elev |
|---|---|---|---|---|---|---|---|---|---|---|---|---|---|---|---|---|---|---|---|---|---|
| Stock | 0.02 | 0.01 | 0.01 | 0.01 | 0.01 | 0.01 | 0 | 0.01 | 0.01 | 0.01 | 0.01 | 0 | 0.01 | 0.01 | 0.02 | 0.02 | 0.01 | 0.01 | 0.02 | 0.02 | 0.02 |
| MAE: default config | 0.02 | 0.01 | 0.01 | 0.01 | 0.01 | 0.01 | 0 | 0.01 | 0.01 | 0.01 | 0.01 | 0 | 0.01 | 0.01 | 0.02 | 0.02 | 0.01 | 0.01 | 0.02 | 0.02 | 0.02 |
| MMCR: global-only | 0.1 | 0.01 | 0.01 | 0.11 | 0.09 | 0.15 | 0.86 | 0.01 | 0.01 | 0.08 | 0.83 | 0 | 0.01 | 0.01 | 0.02 | 0.03 | 0.02 | 0.01 | 0.02 | 0.02 | 0.02 |
| MMCR: global+local | 0.76 | 0.89 | 0.92 | 0.79 | 0.79 | 0.71 | 0.11 | 0.85 | 0.88 | 0.87 | 0.12 | 0.02 | 0.91 | 0.9 | 0.8 | 0.02 | 0.11 | 0.01 | 0.71 | 0.77 | 0.78 |
| MoCoV3: $\tau$=0.1 | 0.04 | 0.03 | 0.02 | 0.04 | 0.05 | 0.06 | 0.01 | 0.05 | 0.07 | 0.02 | 0.02 | 0.02 | 0.03 | 0.03 | 0.1 | 0.11 | 0.82 | 0.87 | 0.13 | 0.1 | 0.09 |
| MoCoV3: $\tau$=1.0 | 0.06 | 0.06 | 0.04 | 0.03 | 0.04 | 0.05 | 0.01 | 0.06 | 0.03 | 0.02 | 0.02 | 0.95 | 0.03 | 0.04 | 0.04 | 0.8 | 0.03 | 0.1 | 0.09 | 0.08 | 0.07 |

*Table 9.* Few-shot comparison across training set sizes ($N$). Results are top-1 % accuracy.

| | Caltech101 | CIFAR-10 | CIFAR-100 | DTD | Flowers102 | Pets | Sun397 | SVHN | Camelyon | EuroSAT | Resisc45 | Retinopathy | Clevr-Count | Clevr-Dist | DMLab | dSpr-Loc-X | dSpr-Loc-Y | dSpr-Ori | KITTI-Dist | sNORB-Azim | sNORB-Elev |
|---|---|---|---|---|---|---|---|---|---|---|---|---|---|---|---|---|---|---|---|---|---|
| **$N = 25$** | | | | | | | | | | | | | | | | | | | | | |
| Stock | 3.3 | 15.9 | 1.2 | 6.8 | 3.3 | 4.1 | 0.3 | 9.8 | 61.5 | 21.0 | 10.1 | 22.5 | 13.3 | 14.1 | 19.4 | 4.0 | 5.0 | 6.9 | 42.9 | 6.3 | 15.3 |
| Uniform Soup | 5.0 | 9.3 | 1.9 | 4.1 | 5.2 | 3.5 | 0.5 | 10.2 | 50.1 | 13.6 | 6.0 | 31.3 | 18.4 | 14.3 | 16.3 | 3.9 | 5.1 | 2.4 | 12.4 | 6.1 | 12.4 |
| Self-Seasoning | 20.1 | 54.3 | 7.1 | 25.0 | 17.8 | 35.1 | 2.5 | 38.7 | 72.2 | 77.7 | 27.6 | 30.1 | 18.0 | 20.2 | 33.1 | 3.3 | 2.5 | 9.9 | 42.8 | 9.2 | 15.5 |
| Labeled Seasoning | 17.0 | 55.2 | 4.6 | 16.8 | 9.5 | 25.4 | 2.1 | 31.8 | 78.0 | 84.9 | 21.3 | 44.7 | 16.2 | 18.5 | 31.3 | 3.0 | 4.6 | 10.3 | 48.2 | 6.8 | 17.3 |
| Fine-tune | 14.9 | 34.2 | 4.3 | 17.2 | 10.4 | 16.9 | 1.9 | 17.4 | 76.4 | 71.2 | 17.3 | 47.9 | 31.3 | 29.5 | 27.5 | 3.8 | 6.6 | 7.5 | 56.3 | 8.3 | 14.4 |
| **$N = 50$** | | | | | | | | | | | | | | | | | | | | | |
| Stock | 10.0 | 20.2 | 2.0 | 3.4 | 4.6 | 3.6 | 1.1 | 10.5 | 68.7 | 35.4 | 8.3 | 9.6 | 19.2 | 19.5 | 12.2 | 3.5 | 5.3 | 5.5 | 36.4 | 6.3 | 14.8 |
| Uniform Soup | 5.4 | 20.1 | 1.7 | 4.4 | 5.3 | 3.0 | 1.0 | 7.4 | 65.2 | 36.6 | 6.1 | 13.2 | 18.8 | 22.6 | 16.6 | 5.5 | 4.8 | 2.9 | 37.1 | 5.9 | 12.9 |
| Self-Seasoning | 45.5 | 62.3 | 9.7 | 33.0 | 32.3 | 46.0 | 4.7 | 56.6 | 74.7 | 77.8 | 40.1 | 38.7 | 16.0 | 22.1 | 28.1 | 2.5 | 3.1 | 19.4 | 49.5 | 10.0 | 15.8 |
| Labeled Seasoning | 35.4 | 62.1 | 6.7 | 30.8 | 16.3 | 44.9 | 2.9 | 56.7 | 77.3 | 85.2 | 51.2 | 21.0 | 18.1 | 20.3 | 28.7 | 3.7 | 5.6 | 13.0 | 52.6 | 10.2 | 15.6 |
| Fine-tune | 34.0 | 43.9 | 5.4 | 21.9 | 16.9 | 21.7 | 2.7 | 21.0 | 80.5 | 72.3 | 28.1 | 35.6 | 37.8 | 30.1 | 27.6 | 3.4 | 11.3 | 7.7 | 57.1 | 9.7 | 18.2 |
| **$N = 75$** | | | | | | | | | | | | | | | | | | | | | |
| Stock | 12.9 | 18.5 | 2.8 | 6.9 | 6.1 | 3.4 | 1.3 | 10.7 | 66.5 | 32.4 | 10.5 | 30.2 | 18.5 | 17.6 | 17.6 | 4.5 | 6.7 | 5.4 | 39.2 | 4.9 | 15.3 |
| Uniform Soup | 7.6 | 19.5 | 3.1 | 3.3 | 7.5 | 4.6 | 0.6 | 9.0 | 67.2 | 33.6 | 7.8 | 23.7 | 23.8 | 19.1 | 16.3 | 5.3 | 4.0 | 2.4 | 32.2 | 5.2 | 14.4 |
| Self-Seasoning | 54.1 | 64.2 | 15.8 | 34.3 | 42.8 | 50.9 | 6.3 | 61.3 | 75.7 | 87.9 | 43.2 | 29.7 | 17.1 | 26.7 | 29.7 | 3.6 | 3.5 | 20.0 | 43.9 | 10.8 | 13.8 |
| Labeled Seasoning | 42.7 | 66.8 | 7.0 | 35.1 | 21.3 | 50.6 | 3.5 | 61.5 | 77.7 | 87.9 | 51.3 | 36.6 | 21.9 | 22.4 | 29.2 | 3.5 | 3.7 | 19.3 | 43.6 | 10.4 | 16.9 |
| Fine-tune | 37.8 | 54.3 | 6.5 | 29.0 | 25.0 | 41.0 | 4.2 | 41.2 | 79.0 | 78.6 | 30.1 | 40.2 | 34.1 | 41.5 | 32.2 | 6.0 | 13.3 | 15.3 | 64.7 | 11.1 | 17.8 |
| **$N = 100$** | | | | | | | | | | | | | | | | | | | | | |
| Stock | 16.7 | 24.3 | 3.6 | 8.1 | 7.0 | 3.6 | 1.4 | 13.2 | 67.4 | 42.4 | 16.3 | 24.3 | 17.0 | 18.2 | 14.6 | 5.4 | 6.3 | 7.3 | 38.3 | 6.1 | 15.1 |
| Uniform Soup | 6.4 | 20.4 | 2.1 | 7.5 | 8.3 | 4.2 | 1.0 | 13.5 | 71.7 | 35.4 | 9.4 | 20.2 | 23.5 | 20.8 | 16.6 | 7.8 | 4.7 | 2.3 | 31.4 | 5.7 | 16.8 |
| Self-Seasoning | 77.0 | 69.3 | 18.9 | 40.4 | 54.3 | 55.0 | 8.2 | 62.2 | 76.7 | 90.8 | 54.3 | 30.8 | 17.4 | 24.1 | 21.1 | 3.5 | 3.6 | 24.6 | 40.2 | 10.4 | 14.9 |
| Labeled Seasoning | 60.0 | 70.8 | 8.5 | 39.2 | 28.2 | 53.2 | 4.9 | 63.1 | 81.0 | 91.5 | 54.7 | 30.1 | 22.7 | 22.5 | 28.8 | 4.2 | 6.1 | 21.2 | 48.2 | 8.6 | 18.6 |
| Fine-tune | 64.6 | 56.2 | 8.3 | 35.2 | 30.2 | 53.8 | 4.4 | 26.4 | 78.2 | 82.2 | 39.3 | 38.7 | 48.7 | 39.4 | 29.2 | 5.8 | 12.6 | 15.6 | 64.0 | 10.9 | 21.3 |
| **$N = 125$** | | | | | | | | | | | | | | | | | | | | | |
| Stock | 6.0 | 20.5 | 2.6 | 9.0 | 5.8 | 5.0 | 1.1 | 10.6 | 71.7 | 45.9 | 17.3 | 31.9 | 16.0 | 21.0 | 20.0 | 6.0 | 5.7 | 7.7 | 46.1 | 7.3 | 18.5 |
| Uniform Soup | 3.8 | 18.4 | 2.9 | 7.2 | 4.7 | 4.1 | 1.0 | 10.3 | 70.8 | 31.2 | 11.7 | 18.1 | 24.0 | 20.7 | 17.5 | 7.8 | 3.6 | 2.9 | 27.8 | 6.6 | 16.5 |
| Self-Seasoning | 78.4 | 70.9 | 22.7 | 45.4 | 56.8 | 53.4 | 9.4 | 68.1 | 78.6 | 92.5 | 55.2 | 28.7 | 16.5 | 27.4 | 22.8 | 4.4 | 4.1 | 26.3 | 39.2 | 9.5 | 17.0 |
| Labeled Seasoning | 77.9 | 71.7 | 22.6 | 43.7 | 48.2 | 53.1 | 5.3 | 68.3 | 80.5 | 92.8 | 56.1 | 33.1 | 25.5 | 24.4 | 25.7 | 4.4 | 4.6 | 26.9 | 49.6 | 11.3 | 18.4 |
| Fine-tune | 51.1 | 59.7 | 7.6 | 38.3 | 35.5 | 51.9 | 4.6 | 54.9 | 81.1 | 87.5 | 41.8 | 60.6 | 55.7 | 43.7 | 30.6 | 5.9 | 13.4 | 16.3 | 67.5 | 10.0 | 21.3 |
| **$N = 150$** | | | | | | | | | | | | | | | | | | | | | |
| Stock | 12.2 | 22.3 | 3.8 | 7.9 | 5.3 | 3.8 | 1.4 | 8.4 | 66.2 | 45.7 | 18.2 | 33.2 | 18.6 | 20.4 | 19.3 | 4.2 | 7.1 | 7.7 | 48.0 | 8.4 | 16.4 |
| Uniform Soup | 3.4 | 22.9 | 2.5 | 7.4 | 7.1 | 3.8 | 1.2 | 10.3 | 74.2 | 44.6 | 11.8 | 39.1 | 23.7 | 18.9 | 18.6 | 6.9 | 5.3 | 2.4 | 32.6 | 7.4 | 16.6 |
| Self-Seasoning | 81.3 | 71.5 | 20.7 | 48.2 | 61.0 | 58.6 | 9.6 | 67.6 | 73.0 | 92.0 | 63.8 | 36.1 | 17.7 | 24.7 | 20.3 | 3.2 | 3.6 | 25.8 | 40.1 | 8.3 | 16.6 |
| Labeled Seasoning | 81.5 | 72.2 | 21.0 | 48.0 | 59.2 | 57.8 | 5.9 | 67.8 | 78.6 | 92.4 | 65.1 | 35.6 | 26.9 | 25.3 | 31.9 | 5.2 | 5.3 | 24.0 | 39.7 | 7.9 | 17.0 |
| Fine-tune | 56.5 | 63.2 | 9.6 | 40.6 | 45.9 | 55.0 | 5.4 | 46.3 | 77.5 | 84.7 | 50.7 | 69.9 | 56.0 | 44.3 | 35.0 | 6.4 | 13.2 | 22.0 | 62.4 | 10.8 | 24.8 |
| **$N = 200$** | | | | | | | | | | | | | | | | | | | | | |
| Stock | 14.4 | 24.4 | 3.4 | 10.1 | 6.6 | 5.0 | 1.6 | 10.0 | 71.7 | 50.3 | 17.2 | 36.3 | 19.9 | 21.2 | 18.6 | 4.0 | 6.4 | 8.1 | 45.1 | 8.2 | 15.9 |
| Uniform Soup | 6.5 | 22.3 | 3.1 | 7.2 | 6.0 | 4.5 | 1.6 | 13.3 | 73.7 | 46.3 | 11.2 | 35.9 | 28.1 | 23.4 | 18.9 | 9.2 | 5.1 | 2.3 | 33.8 | 6.1 | 14.6 |
| Self-Seasoning | 82.6 | 73.1 | 22.1 | 50.3 | 64.9 | 58.6 | 9.6 | 70.6 | 80.7 | 91.8 | 63.7 | 30.4 | 19.5 | 25.6 | 19.4 | 4.1 | 3.8 | 25.9 | 43.9 | 8.8 | 17.6 |
| Labeled Seasoning | 81.9 | 73.1 | 22.8 | 48.7 | 62.6 | 57.4 | 7.1 | 71.0 | 82.4 | 92.4 | 63.8 | 34.4 | 25.4 | 25.7 | 32.3 | 5.9 | 5.3 | 26.5 | 48.2 | 10.3 | 19.5 |
| Fine-tune | 65.5 | 65.0 | 13.5 | 42.3 | 52.4 | 57.9 | 5.8 | 63.1 | 81.7 | 86.7 | 53.3 | 42.1 | 57.1 | 40.5 | 38.8 | 5.7 | 13.5 | 19.1 | 76.8 | 10.5 | 25.8 |
| **$N = 300$** | | | | | | | | | | | | | | | | | | | | | |
| Stock | 12.9 | 23.6 | 4.4 | 10.9 | 8.9 | 5.9 | 2.5 | 10.7 | 71.5 | 54.9 | 23.1 | 28.8 | 20.6 | 25.9 | 17.8 | 5.0 | 8.9 | 9.2 | 43.7 | 9.8 | 18.4 |
| Uniform Soup | 6.5 | 23.0 | 3.2 | 6.7 | 8.8 | 5.2 | 1.0 | 9.8 | 72.5 | 48.7 | 14.6 | 37.3 | 24.5 | 28.4 | 17.8 | 5.7 | 9.1 | 4.6 | 36.8 | 5.0 | 20.1 |
| Self-Seasoning | 87.5 | 74.7 | 26.4 | 52.9 | 68.1 | 64.2 | 14.5 | 76.0 | 77.7 | 93.8 | 64.9 | 33.9 | 17.1 | 29.7 | 22.5 | 4.3 | 4.7 | 29.9 | 41.9 | 10.4 | 18.7 |
| Labeled Seasoning | 86.5 | 75.8 | 30.3 | 52.1 | 68.9 | 65.0 | 9.6 | 75.2 | 79.3 | 93.4 | 66.2 | 34.7 | 27.5 | 26.2 | 31.0 | 3.8 | 7.2 | 24.9 | 41.2 | 11.4 | 21.5 |
| Fine-tune | 67.4 | 74.6 | 11.8 | 49.6 | 61.6 | 79.4 | 7.4 | 72.3 | 79.8 | 91.0 | 60.3 | 68.5 | 68.9 | 49.3 | 39.7 | 7.2 | 21.0 | 24.7 | 75.2 | 12.7 | 27.0 |
| **$N = 500$** | | | | | | | | | | | | | | | | | | | | | |
| Stock | 18.8 | 26.1 | 4.5 | 13.9 | 9.8 | 7.0 | 2.2 | 11.0 | 73.7 | 61.8 | 26.1 | 46.9 | 23.6 | 23.2 | 20.7 | 4.7 | 8.8 | 14.1 | 44.0 | 9.4 | 24.0 |
| Uniform Soup | 7.8 | 25.9 | 5.8 | 9.3 | 11.7 | 7.4 | 1.3 | 10.6 | 73.2 | 51.3 | 19.1 | 40.3 | 27.1 | 24.2 | 19.9 | 9.4 | 10.2 | 4.8 | 36.7 | 6.3 | 20.4 |
| Self-Seasoning | 86.2 | 76.7 | 30.3 | 55.8 | 73.6 | 67.9 | 17.4 | 76.9 | 78.8 | 94.6 | 71.1 | 48.5 | 17.3 | 28.0 | 23.6 | 4.2 | 7.0 | 35.7 | 50.4 | 8.6 | 18.6 |
| Labeled Seasoning | 86.3 | 78.3 | 32.2 | 55.5 | 72.2 | 67.7 | 18.0 | 76.6 | 81.1 | 94.6 | 71.2 | 54.6 | 29.1 | 26.8 | 29.4 | 3.4 | 6.4 | 34.0 | 49.6 | 10.5 | 24.2 |
| Fine-tune | 83.6 | 81.3 | 27.1 | 57.0 | 74.6 | 81.6 | 9.9 | 76.1 | 83.3 | 93.6 | 71.2 | 72.8 | 74.4 | 53.9 | 44.2 | 8.5 | 25.7 | 32.8 | 81.0 | 10.9 | 33.7 |
| **$N = 1000$** | | | | | | | | | | | | | | | | | | | | | |
| Stock | 27.2 | 30.0 | 6.9 | 17.7 | 13.7 | 7.6 | 3.0 | 15.3 | 74.1 | 68.7 | 29.7 | 56.7 | 25.5 | 27.2 | 22.5 | 6.5 | 10.8 | 20.5 | 50.4 | 11.5 | 24.1 |
| Uniform Soup | 25.8 | 25.8 | 4.7 | 16.6 | 10.8 | 5.0 | 1.9 | 12.1 | 74.0 | 58.5 | 22.8 | 55.2 | 27.4 | 25.5 | 21.2 | 7.8 | 8.9 | 6.9 | 39.8 | 6.3 | 28.8 |
| Self-Seasoning | 88.6 | 78.9 | 35.2 | 62.7 | 78.1 | 71.9 | 19.9 | 78.4 | 81.7 | 95.2 | 74.6 | 60.6 | 17.1 | 30.6 | 27.1 | 5.0 | 8.2 | 37.2 | 47.8 | 9.8 | 20.3 |
| Labeled Seasoning | 88.1 | 80.8 | 36.1 | 61.7 | 78.2 | 70.6 | 23.3 | 78.4 | 81.2 | 95.3 | 74.7 | 62.0 | 31.0 | 30.8 | 35.5 | 11.1 | 7.3 | 37.2 | 51.2 | 11.8 | 24.0 |
| Fine-tune | 91.2 | 89.6 | 37.5 | 64.2 | 88.3 | 86.9 | 31.2 | 84.4 | 85.9 | 95.7 | 79.9 | 75.3 | 86.9 | 59.5 | 54.3 | 13.9 | 28.6 | 41.9 | 81.4 | 16.7 | 39.7 |

