# OpenReview forum: "Self-Soupervision: Cooking Model Soups without Labels"
_ICML.cc/2026/Conference — ICML 2026 spotlight_

### Official Review · Reviewer_4UN5 · 2026-03-03

**Soundness:** 4
**Presentation:** 3
**Significance:** 3
**Originality:** 3
**Overall Recommendation:** 5
**Confidence:** 4

**Summary:**

This work proposed Self-Soupervision, a framework for model soup using self-supervised learning (SSL). The core technical contribution is using SSL losses to create various ingredients for different without any unlabeled data, different from prior work in supervised learning manner. Experiments have demonstrated its competitive benefits over supervised learning.

**Compliance With Llm Reviewing Policy:**

Affirmed.

**Final Justification:**

This work advances self-supervised learning (SSL) in the context of model souping. The presentation is in high quality. Extensive experiments have been conducted to demonstrate its advantages. The authors further resolved my clarification questions in the rebuttal. I raised my rating to __5: Accept__.

**Key Questions For Authors:**

I think this paper has merits and deserves a high score, but I would like to hear about the clarification concerns mentioned in the Weaknesses above before raising my score.

**Limitations:**

For technical limitations, yes. But negative societal impact is not discussed. Despite more ingredients can improve accuracy, one potential impact could be energy consumption proportional to the size of ingredients and each of their fine-tuning, compared to one single fine-tuning.

**Strengths And Weaknesses:**

__Soundness (SN)__

__Strengths__:

__SN_S1__: The visualization in Figure 2 is presented in high quality and demonstrates clearly that mixing various ingredients by SSL can improve performance in green triangles.

__SN_S2__: Overall, Table 1 shows the proposed “Self-Soup” improves accuracy over the other two baselines on ImageNet.

__SN_S3__: This work has conducted several more experiments which clear logic flow from various aspects, including Table 2: the gain of Self-Soupervision on the test distribution, effective test-time adaptation shown in Table 2, transfer ability and robustness in Table 4, interesting Self-Souping results in Figure 4 and its advantage over supervised seasoning in Table 5.

---

__Presentation (P)__

__Strengths__:

__P_S1__: This work is written in high quality with compact information, clear figures and smooth logical flow. I enjoy reading this paper and is able to follow the logic easily.

__P_S2__: (Minor) The paper uses some humor to describe the technical aspects especially when it comes to ingredients and soup. Examples include Line 18 (right) “depriving our palettes of tasty new soups for many occasions” and Line 251 (right) “a novel finding that whets the appetite for more tasty soups now that it is possible.”

__Weaknesses__:

__P_W1__: (Clarity) The term “inter-train” is never formally defined. What does “inter-train” models mean in terms of training, test set, label and unlabeled data? Does it mean training models while swapping their datasets/subsets that are different from the one where the stock is pre-trained on? Such clarity is needed.

For example, it would be great to expand to details on Line 267 “we first inter-train 3 models on the ImageNet-1K training set for 5 epochs with a 1e-5 learning rate.” in terms of dataset.

__P_W2__: (Clarity) In Figure 1 and 3, do circles in different horizontal layers represent exactly the same dataset throughout the process from top stock to down final model or soup?

__P_W3__: (Clarity) In general, the term “Train” has a broad meaning as it refers to any procedure that relates to model weight updates. But in Figure 1 and 3 in this model soup context, it (“Train”) seems to refer to either “fine-tune” (when labels are available in supervised learning) and "train using SSL”. I am not trying to change the term, just ask for clarity if this understanding is correct.

__P_W4__: (Clarity) Line 217 states “sample the M mixture coefficients uniformly from the probability simplex.” I am trying to understand where the probability comes from. It would be great to provide more clarity in terms of seasoning mixes, its relation to classifiers (?) in k tasks, the stock, ingredients and SSL.

__P_W5__: (Clarity) In Table 1, what criterion is used for Greedy Search? Is it based on accuracy in the validation set?

__P_W6__: The manuscript appears to have been heavily polished by an LLM. Using an LLM to assist in writing is allowed as stated by the author policy, but the fact that the excessive use of em bashes (—) found in 18 places in its current form suggests that a large portion of this paper may have been polished by an LLM. Em dashes are the typical features that an LLM tends to generate especially ChatGPT as they appear in English literature. In contrast, em dashes are much sparser in academic papers especially prior to the LLM age. I would like to point it out to raise awareness. Specifically, em dashes are found in the following places: Line 18 (right), 30 (left), 49 (right), 156 (left), 160 (right), 182, 184, 198 (left), 212 (left), 222, 250 (right), 260 (right), 321 (left), 378 (right), 382 (left), 380 (right), 386 and 434 (right).

---

__Significance (SF)__

__Strengths__:

__SF_S1__: This work explores Self-Souping in which model soups are made by SSL without labels compared to prior work in supervised fashion. This is an important direction that aims at reducing the reliance of labor-intensive labels.

__SF_S2__: Despite its simplicity, the PyTorch code is presented .

---

__Originality (OR)__

__Strengths__:

__OR_S1__: Exploring self-supervised learning (SSL) in the context of model souping is a novel direction.

---

> ### Author Rebuttal · Authors · 2026-03-31
>
> We thank the reviewer for the attention to detail and precise questions about the terminology, method, and results.
>
> > [P_W1] “inter-train” is never formally defined. What does “inter-train” models mean in terms of training, test set, label and unlabeled data?
>
> Thank you for highlighting this: we will edit Sec. 1 and 2 to define this term vs. fine-tuning and training more generally. Inter-traing is the intermediate stage(s) of optimization between the pre-training of the stock and the fine-tuning of the final task model. It is not defined by train/test split or labeled/unlabeled data. We intend to inherit the definition from Model Ratatouille (Ramé et al. 2023) but will make this more self-contained.
>
> > [P_W2] In Figure 1 and 3, do circles in different horizontal layers represent exactly the same dataset
>
> Each circle is a model checkpoint and not a dataset. The datasets at each step depend on the method (Ratatouille requires different datasets while Self-Soupervision can use the same dataset or _optionally_ different data) and on the setting (Self-Soupervision can do SSL on unlabeled data like a shifted target test set then supervised learning on different labeled data).
>
> To clarify this we will edit the captions of Fig. 1 and 3 and add different lighter/darker arrows to Ratatouille to indicate inter-training on different datasets.
>
> > [P_W3] “Train” has a broad meaning as it refers to any procedure that relates to model weight updates
> > just ask for clarity
>
> Yes, in this work we use "train" in this general sense as shorthand for an optimization that given a choice of dataset and weights results in updated weights. We will edit Sec. 2 to review the terminology as part of the background.
>
> > [P_W4] sample the M mixture coefficients uniformly from the probability simplex
> > where the probability comes from
>
> The mixture coefficients are constrained to sum to one as a convex combination. In this sense, the contribution of each ingredient is normalized, and so we refer to it as a probability. Furthermore, we do so to use the Dirichlet distribution to sample mixtures from the probability simplex over M elements.
>
> > [P_W5] what criterion is used for Greedy Search? [is it validation accuracy?]
>
> Yes, please see the caption of Tab. 1 "For greedy search and best ingredient, we select based on IN-Val
> accuracy". Note this means that the "greedy" result and "best" ingredient can therefore be worse on other evaluations than IN-Val as seen in the results.
>
>
> > negative societal impact is not discussed. Despite more ingredients can improve accuracy, one potential impact could be energy consumption
>
> In our impact statement we discuss the counterpoint, that a fine-tuning for souping may be more computationally efficient than training/pre-training a custom model from scratch. However, we recognize that encouraging souping could result in more fine-tuning overall, so we will incorporate this point into our discussion and impact statement to reflect this diversity of opinion and potential outcomes. We thank the reviewer for consider the possible effects.
>
> > P_W6: The manuscript appears to have been heavily polished by an LLM.
> > excessive use of em bashes (—)
>
> We respectfully note that the writing was done entirely manually without any LLM editing. The use of dashes is a stylistic choice.
>
> Please let us know if we can further clarify any points or terminology in the discussion. We will incorporate the suggestions about where and when definitions are needed into our revisions.

---

> > ### Author Rebuttal · Reviewer_4UN5 · 2026-04-01
> >
> > This work is presented in high quality. The main concerns were mainly about clarification and the authors have provided further clarity in the rebuttal.

---

> > > ### Author Response · Authors · 2026-04-06
> > >
> > > Thank you for your time and attention in reviewing this work and engaging in the discussion.

---

### Official Review · Reviewer_Qgjq · 2026-03-12

**Soundness:** 3
**Presentation:** 4
**Significance:** 3
**Originality:** 3
**Overall Recommendation:** 5
**Confidence:** 3

**Summary:**

This paper extends model soups to self-supervised learning, termed Self-Soupervision. Specifically, the authors propose to inter-train the stock model using various SSL algorithms on unlabeled data, typically from test or shifted distributions (e.g., corrupted data), before possible subsequent supervised finetunings. To facilitate model mixing, the authors also introduce Self-Seasoning that optimizes mixing weights by minimizing the entropy of nearest-neighbor predictions, rather than using labeled validation sets. Experiments verify that ingredients from different SSL methods and parameters are able to produce effective model soups, and can help transfer to unlabeled target distribution during training time.

**Compliance With Llm Reviewing Policy:**

Affirmed.

**Final Justification:**

The authors have successfully provided a convincing comparison with supervised soups. These updates have significantly strengthened the paper's empirical positioning. Based on the high quality of the submission, I raise my score to 5:accept.

**Key Questions For Authors:**

1. Does the proposed self-soupervision's inter-training also involve different data like in Model Ratatouille?
2. What seasoning is used in this paper, when: 1) the supervised finetuning is dropped, and 2) the supervised finetuning is kept? If the seasoning for 2) is the proposed self-seasoning, does it perform better than the supervised seasoning?
3. It seems the supervised soup in comparison only uses $N$ ingredients but not $MN$ ones. This may induce unfairness when compared to the self-soup implementation. When compared in an equal-computation setting ($MN$ ingredients), does the proposed method beat fully supervised soups?

**Limitations:**

Currently, the paper is unclear whether the SSL model soup can perform better than the supervised model soup in a fair and pure setting (see weaknesses and questions). The self-seasoning is novel but yet limited in performance.

**Strengths And Weaknesses:**

Strengths:

The main idea of extending model soups to self-supervised learning schemes is highly novel. The experimental finding that models trained with fundamentally different SSL methods satisfy the LMC condition is valuable. Meanwhile, the method is empirically shown to be useful for transfer learning, opening a new research path. For presentation, the paper is quite well-written.

Weaknesses:

- The effectiveness in Sec. 4.1 may come from supervised model soup + stochasticity (brought by few-epoch SSL), as it is also stated in Sec. 2.2 that supervised finetuning with stochasticity is a standard effective model soup. Here, it lacks an experiment to discard the supervised part completely to prove the effectiveness.
- The numbers presented in Sec. 4.4 is confusingly unaligned with Table 4. Meanwhile in the results, the supervised model soups seem to be even better in substantial cases.
- From results in table 5, the model produced by the proposed self-seasoning is usually (15 out of 21 datasets) worse than its best ingredient, sometimes even not competent to the stock model. This degrades the contribution of self-seasoning in the paper.

Minors:
- In Sec.4's description, the continued SSL mixes several fine-tunings that originate from one inter-training. This is inconsistent to Figure 3.
- Table 5 is not referenced in the text.

---

> ### Author Rebuttal · Authors · 2026-03-31
>
> We thank the reviewer for the careful questions about self-soupervision and controlled experiments vs. supervised soups.
>
> > effectiveness in Sec. 4.1 may come from supervised model soup + stochasticity
> > it lacks an experiment to discard the supervised part completely to prove the effectiveness
>
> Thank you for the suggested experiment to provide stronger evidence for Self-Soupervison. We experiment with different numbers of ingredients. A smaller self-soup with four ingredients rivals a larger supervised soup with eight ingredients and a larger self-soup with eight ingredients is better still.
>
> - supervised soup with 8 ingredients (4 LRs and 2 augmentation types)
> IN-Val=79.14%, IN-C=32.95%
> - self-soup with 4 ingredients (4 SSL algos = MAE, MoCoV3, MMCR, LeJEPA):
> IN-Val=79.37%, IN-C=32.26%
> - self-soup with 8 ingredients (4 SSL algos and 2 augmentation types)
> IN-Val=79.33%, IN-C=33.91%
>
> Even when the supervised soup has more ingredients (mixing stochasticity, learning rates, augmentation types = RandAug and 3Aug, and random seeds), our self-soup (mixing only different self-supervised learning methods) achieves near equal or better accuracy in-distribution (IN-Val) and out-of-distribution (IN-C).
>
> > The numbers presented in Sec. 4.4 are confusingly unaligned with Table 4.
>
>
> We politely ask for clarification of which numbers do not align? For instance, the 67.5 of Self-Soup with Greedy Search is the green bold result in the leftmost column of Tab. 4 (b). To help navigate these tables, we will edit Sec. 4.4 to refer to Table 4 (a) for the results across datasets and Table 4 (b) for the results across shifts.
>
> > supervised model soups seem to be even better in substantial cases [in Table 4]
>
>
> Yes. As we mention in the caption of Tab. 4 (a), self-soups are best at the "natural" tasks. Supervised soups are better at the "structured" tasks. In future work more mixing of more ingredients could improve across all the datasets of VTAB, but these soups still need cooking.
>
> > the model produced by the proposed self-seasoning is usually (15 out of 21 datasets) worse than its best ingredient, sometimes even not competent to the stock model.
>
> Please note self-seasoning is _unsupervised / label-free_ while the best ingredient is informed by labels. Self-seasoning is only worse than the stock in 1 out of 21 datasets. We do not claim it is the most accurate, but contribute it as the first unsupervised method for seasoning.
>
> > [In Sec. 4] continued SSL mixes several fine-tunings that originate from one inter-training. This is inconsistent to Figure 3.
>
> Sec. 4 and Fig. 3 are indeed separate: the Continued SSL baseline mixes a soup while Fig. 3 illustrates the equal computation and tuning of model selection vs. model souping. We will edit Sec. 4 to reference Fig. 3 as an illustration of computational cost rather than our baseline.
>
> > Does the proposed self-soupervision's inter-training also involve different data like in Model Ratatouille?
>
> No, and this is a key advantage of Self-Soupervision: while Model Ratatouille requires different data and labeled data, Self-Soupervision can be applied to the same data and unlabeled data.
>
> Please see the response to uxP6 for rebuttal results about Model Ratatouille: Self-Soupervision is equally good without inter-training on auxiliary datasets and labels.
>
> > What seasoning is used in this paper, when: 1) the supervised finetuning is dropped, and 2) the supervised finetuning is kept?
>
> To clarify, seasoning is an alternative to supervised fine-tuning: all the results of Tab. 5 are without supervised fine-tuning. With labels we do seasoning to minimize cross entropy by picking the best random mixture, and without labels we do self-seasoning to minimize entropy of predictions. In both cases the predictions are derived by k-NN between the representations extracted of labeled train and unlabeled test points.
>
> > When compared in an equal-computation setting (M*N ingredients), does the proposed method beat fully supervised soups?
>
> At equal computation a self-soup improves on a supervised soup for a soup of 8 ingredients. Inter-training across different SSL losses may offer more diversity for souping improvement than optimization hyperparameters and stochasticity alone.
>
> - supervised soup with 8 ingredients (4 LRs and 2 aug types)
> IN-Val: 79.14% IN-C: 32.95%
> - self-soup with 4 ingredients (our 4 SSL algos):
> IN-Val: 79.37%	IN-C: 32.26%
> - self-soup with 8 ingredients (4 SSL algos and 2 aug types)
> IN-Val: 79.33% IN-C: 33.91%
>
> > Currently, the paper is unclear whether the SSL model soup can perform better than the supervised model soup in a fair and pure setting
>
>
> With the new experiments thanks to the reviewer suggestions, we now have evidence that self-soupervision equals or improve on supervised soups _when they are possible_ and offers more options _when they are not possible_ due to lack of labels or different datasets. We are happy to discuss these results or any other details.

---

> > ### Author Rebuttal · Reviewer_Qgjq · 2026-04-01
> >
> > After re-reading the paper, I found the data confusion solved. Importantly, the authors have mostly solved my concerns about comparison with supervised soups. I would raise my score for the paper's good quality.

---

> > > ### Author Response · Authors · 2026-04-06
> > >
> > > Thank you for your time and attention in reviewing this work and engaging in the discussion.

---

### Official Review · Reviewer_JDdS · 2026-03-13

**Soundness:** 3
**Presentation:** 3
**Significance:** 2
**Originality:** 2
**Overall Recommendation:** 4
**Confidence:** 3

**Summary:**

In this paper, the authors study the problem of model souping for models trained by Self-Supervised Learning (SSL) methods. Based on Model Ratatouille, they proposed Self-Soupervision which replaces the first stage of supervised training in Model Ratatouille with SSL. They showed that linear-mode connectivity exists for these models even if they are trained with different SSL methods (ie., .starting with one pretrained SSL model, the mixed weights of models finetuned with different SSL methods yields better performance than any single finetuned model). They also showed that Self-Soupervision yields better performance if it has access to the unlabelled test distribution.

**Compliance With Llm Reviewing Policy:**

Affirmed.

**Final Justification:**

This work explores leveraging self-supervised models for souping, and provides insights for this route. The rebuttal addressed my concerns. The only thing is that the proposed method is similar to a previous work named Model Ratatouille, which limits the impact of this work. I am leaning towards accept but will let the AC decide.

**Key Questions For Authors:**

- Key Questions
  1. How does the average performance across different tasks look like if it is plotted similarly to Figure 2?
  2. What is the setup for Supervised Soup in Section 4.2 (e.g., the number of epochs, the number of ingredients, etc)?
  3. What is the performance of Continued SSL + Supervised Soup if we apply a similar training strategy as Self-Souping in Section 4.3 (ie., first SSL on test distribution, then supervised soup on the training distribution)?
  4. What is the setup for Supervised Soup in Section 4.3?
  5. What is the motivation of designing Self-Soupervision in its current form instead of other ways to incorporate SSL, such as
      1. Finetuning the stock with SSL to obtain different ingredients, then mixing (similar to vanilla Model Soup)
      2. Inter-training on labelled data, then SSL on target distribution (ie., switching the two stages in Self-Soupervision)
- Other Questions
  1. Line 211 left part: 'We instead mix purely self-supervised ingredients into a model for representation' Does this mean the mixing is done after the first stage of Self-Soupervision (ie., after the green arrows in Figures 1 and 3)?
  2. Line 203 left part: 'A continued SSL + supervised soup ... mixes several fine-tunings that originate from one inter-training' Does this mean continued SSL + supervised soup also mixes several models into one soup? This seems to be different from Figure 3, where there is only one dashed arrow.
- Suggestions
  1. Line 190 right part: It would be better to write down the mathematical formulation of 'minimize the entropy of predictions by nearest neighbors'
  2. Line 313 left part: It would be better to add a reference to Table 2 when mentioning the results.
  3. Line 379 left part: It would be better to add a reference to Table 5 when mentioning the results..
- Typos
  1. Line 326 left part and Line 410 right part: (Wang* et al., 2021) -> (Wang et al., 2021)

**Limitations:**

yes

**Strengths And Weaknesses:**

- Soundness: The results in this work are technically sound, and the claims are well supported. The experiments are well-designed to demonstrate the advantage of the proposed method (although a few points are not clearly stated, please see Key Questions)
- Presentation: The paper is well-structured and the presentation is clear. The work clearly discusses prior literature and how the proposed methods differs from existing approaches.
- Significance: This work studies the problem of model souping for models training with SSL methods, which is interesting and significant. The authors provide a practical approach to improve model performance without labels. However, the design choice of the proposed approach seems not well-studied (please see question 5 in Key Questions).
- Originality: The ides of combining model soup with SSL methods is novel, but the proposed approach seems a straightforward extension of Model Ratatouille by replacing one supervised stage with self-supervised training.

---

> ### Author Rebuttal · Authors · 2026-03-31
>
> We thank the reviewer for the detailed questions about our new self-soupervision method, baselines, and alternatives.
>
> > combining model soup with SSL methods is novel
> > a straightforward extension of Model Ratatouille
>
> Model Ratatouille and our Self-Soupervision have a similar structure of staged fine-tuning and souping, as we diagram in Fig. 1, and we respectfully underline the empirical novelty of this change. We report for the first time that linear mode connectivity—as defined in Model Ratatouille—can hold across multiple losses including supervised vs. self-supervised losses and across different self-supervised losses. Furthermore our Self-Soupervision requires neither labels nor auxiliary datasets for its inter-training unlike Ratatouille which requires both.
>
> Please see the response to uxP6 for rebuttal results about Model Ratatouille: Self-Soupervision is equally good without inter-training on auxiliary datasets and labels.
>
> > How does the average performance across different tasks look like if it is plotted similarly to Figure 2?
>
> If we average performance across the test sets, then the plot shows a similar effect. In the revision we can include this average plot, and can also plot relative improvement vs. absolute improvement if that is of interest.
>
> > What is the performance of Continued SSL + Supervised Soup if we apply a similar training strategy as Self-Souping in Section 4.3
>
> Thank you for the suggestion of this additional baseline to complement our baseline of Continued SSL + Supervised Souping (which is itself a new if technically straightforward method): we experiment and self-soup is better at shift:
>
> - Inter-training on IN-C:
> Cont-SSL + Supervised Soup: Val=79.16%, IN-C=35.09%, LAION-C=22.63%
> Self-Soup: Val=78.96%, IN-C=35.72%, LAION-C=23.28%
>
> - Inter-training on LAION-C:
> Cont-SSL + Supervised Soup: Val=79.15%, IN-C=31.65%, LAION-C=27.43%
> Self-Soup: Val=78.87%, IN-C=32.58%, LAION-C=29.48%
>
> > What is the motivation of designing Self-Soupervision [...] instead of other ways
> > Finetuning the stock with SSL to obtain different ingredients, then mixing [mixing pre-supervision]
> > Inter-training on labelled data, then SSL on target distribution [switching stages]
>
> The first option [mixing pre-supervision] is related to seasoning and our self-seasoning: we inter-train the stock by SSL, mix these M ingredients, then make predictions by k-NN over the resulting representation of the mixed model (pg. 4 "Mixing by Instant Seasong"). The second option [switching stages] risks interfering with supervised learning with its final stage of SSL. Our design avoids this risk with its final supervised learning stage.
>
>
> We design Self-Soupervision as a parallel to Model Ratatouille for the study of self-supervision in model soups while keeping everything else we can the same. For this reason we think it is the most informative design.
>
> > Line 211 left [...] Does this mean the mixing is done after the first stage of Self-Soupervision[?]
>
> Yes, this exactly how our "instant seasoning" works and is identical to the alternative suggested by the reviewer [mixing pre-supervision].
>
> > Line 203 left [...] Does this mean continued SSL + supervised soup also mixes several models into one soup?
> > This seems to be different from Figure 3
>
> Yes, except unlike our self-soupervision this baseline does _not mix_ across SSL inter-trainings: it mixes only across supervised fine-tunings as in a standard model soup.
>
> Fig. 3 is indeed different than l. 203  because it shows Continued SSL + Fine-Tuning, which is not a soup, just as a single fine-tuning is not a soup (Fig. 3 right). This is an alternative without any mixing that can be made from the same inter-trainings and fine-tunings as self-soupervision. We will edit Sec. 2.3 ("Continued SSL") and Sec. 4 (pg. 4) to clarify this difference and the purpose of Fig. 3 as illustrating computational cost rather than the baseline.
>
> We compare to Continued SSL + Supervised Souping instead as a stronger baseline than continued SSL + fine-tuning alone.
>
> > Line 190 right [...] the mathematical formulation of 'minimize the entropy of predictions by nearest neighbors'
>
> We will edit to include this equation in appendix Sec. C and point to it from Sec. 3. Please note that Sec. C includes code to explain the calculation of this entropy as a complementary way to describe it.
>
> > Line 313 left [...] reference to Tab. 2
>
> We will edit to reference Tab. 2.
>
> > Line 379 left [...] reference to Tab. 5
>
> We will edit to reference Tab. 5.
>
> > Typos
>
> Thank you for the attention to detail—we include the star because it is a reference with starred first authors but we can remove it to avoid confusion.
>
> Please let us know if we can further discuss the design of Self-Soupervision, the technical details of seasoning, and the experimental results in the submission and this response.

---

> > ### Author Rebuttal · Reviewer_JDdS · 2026-04-04
> >
> > Thank you for your response. My concerns are addressed except the similarity between the proposed method and Model Ratatouille, which limits the contribution to the community. But this work provides insights of using self-supervised models for souping, which are novel. I will raise my score to 4 and let the AC to decide in this case.

---

> > > ### Author Response · Authors · 2026-04-06
> > >
> > > Thank you for your time and attention in reviewing this work and engaging in the discussion.

---

### Official Review · Reviewer_uxP6 · 2026-03-16

**Soundness:** 3
**Presentation:** 2
**Significance:** 2
**Originality:** 3
**Overall Recommendation:** 4
**Confidence:** 3

**Summary:**

The paper propose a unsupervised learning strategy for model soup. It first apply different unsupervised learning to create multiple backbones, then finetune each of it with different ingredients to make the model soup. It shows the effectiveness of unsupervised learning in model soup. The experiment compare with supervised soup and SSL+supervised soup, soup on SSL further improve the results.

**Compliance With Llm Reviewing Policy:**

Affirmed.

**Final Justification:**

This paper provides insights on self-supervised model soup methods. The rebuttal has addressed my major concern on its effectiveness and comparison with ratatouille, so I raised my original score to 4 as the final justification

**Key Questions For Authors:**

see weakness above

**Limitations:**

yes

**Strengths And Weaknesses:**

Strengths
- The method enlarge the pool of model merging, showing that model trained with unsupervised learning can also be merged to boost performance.
- The result shows improvement compared with supervised soup, while it doesn't require as much data

Weakness
- have you compared model ratatouille (in figure 1) with the proposed self-soup.
- The baseline Continued SSL + Fine-tuning sounds like a greedy search version of self-soup. What is the difference between them? And what is being proved by comparing with this baseline?
- section 4 says "they both search over M SSL inter-trainings and N supervised fine-tunings". From figure 3, it should be M*N supervised fine-tunings? since each SSL inter-training requires N supervised fine-tunings
- what is the difference between best ingredient and greedy search
- The Uniform Mix seems the best mixing method, any reason why

---

> ### Author Rebuttal · Authors · 2026-03-31
>
> We thank the reviewer for the questions about the method, baselines, and results that we clarify:
>
> > have you compared model ratatouille
>
> We include Ratatouille in Figure 1 to orient the reader, to explain the structural similarity to Self-Soup with inter-training, and to note the difference in requirements with supervision vs. self-supervision. Self-Soup and Ratatouille both inter-train to first diversify / prepare the ingredients. Then, they fine-tune to a task and mix the ingredients for better generalization than the original model soups, which have only 1 training stage and do not achieve as much ingredient diversity. However, Ratatouille needs several auxiliary labelled datasets for its inter-training. For example, if the target downstream task is ImageNet-1K, Ratatouille needs to inter-train on several _other_ _labelled_ datasets first, then fine-tune on ImageNet-1K and mix the ingredients. Our Self-Soup is more general since it (1) does not need these extra labelled datasets (our ImageNet experiments in Tab. 1 & 2), (2) can harness unlabelled data (our VTAB fine-tuning experiments in Tab. 4a & 4b), and (3) mix without task-fine-tuning for fully unsupervised soups (our VTAB seasoning experiments in Tab. 5).
>
> We now compare to Ratatouille as requested. We inter-train 4 times using the natural tasks from VTAB {Flowers, Pets, Caltech101, and Sun397} by supervised learning. Then for each inter-trained model, we fine-tune 4 times using learning rates {6e-8, 8e-8, 1e-4, 1.5e-4} and uniformly mix all 16 ingredients. We compare this soup with our Self-Soup that we first inter-train 4 times using SSL algorithms {MAE, MoCoV3, MMCR, LeJEPA}, then fine-tune each 4 times using the same learning rates and uniformly mix all 16 ingredients. They perform the same:
>
> - Ratatouille (uniform soup over Flowers+Pets+Sun397+Caltech101):
> IN-Val=79.05%, IN-C=32.20%, LAION-C=22.89%
>
> - Self-Soup (uniform soup over LeJEPA+MAE+MMCR+MoCoV3):
> IN-Val=79.09%, IN-C=32.34%, LAION-C=22.92%
>
> This shows Self-Souping can match Ratatouille on accuracy, with less requirements, which is a practical improvement.
>
> > Continued SSL + Fine-tuning [...] What is the difference [and] what is being proved by comparing with this baseline?
>
> Continued SSL + fine-tuning or supervised souping optimizes the _same_ SSL loss while our Self-Soup optimizes _different_ SSL losses. We are the first to show souping from SSL to supervised losses, in both the continued SSL baseline and in the more significant switch from one SSL loss to another in our Self-Soup method. Our experiments are the first to establish that LMC can hold across fine-tuning to different losses, in our case self-supervised losses, so that souping across SSL methods is both possible (accuracy is maintained) and productive (accuracy even improves).
>
> > they both search over M SSL inter-trainings and N supervised fine-tunings
> > should be M*N supervised fine-tunings?
>
> It is indeed N * M, which is what we intended to express by M and N, but we now realize this could be interpreted as the sum and not the product. We will edit this line in Sec. 4 to add "(for N * M optimizations)" to disambiguate and confirm the interpretation of Fig. 3.
>
> > difference between best ingredient and greedy search
>
> The best ingredient simply selects the one best model by the validation metric, while the greedy search selects the best set of models when souped by starting with the one best model then adding each model in turn if it improves the validation metric. Best ingredient is a simple baseline of our own design, to sanity check mixing, and greedy search is a mixing method due to the original soups (Wortsmann et al. 2022).
>
> > Uniform Mix seems the best mixing method, any reason why
>
> Thank you for this question. It is due to the tuning and how the best ingredient and greedy search require labels. it is only possible to validate on the labeled data, so best and greedy are tuned to the in-distribution validation set of ImageNet, which is  prone to mis-selecting a mix that is better on the training data than testing data when train and test differ due to shift (as for IN-C, IN-R, … in Tab. 1).
>
> Please see the caption of Tab. 1: _For “greedy search” and “best ingredient”, we select based on IN-Val
> accuracy—thus, greedy may not always rank first for every test set._  Based on these two questions about the best ingredient, greedy search, and uniform mix, we will add a paragraph on "Mixing" before Sec. 4.1.
>
> Note that the positive results for the uniform mix are a practical strength: the uniform mix is simple, quick, and needs no tuning.
> For further customization and improvement we can apply seasoning (based on few-shot labels for validation) and our new self-seasoning (based on entropy minimization for _unsupervised_ validation) as we do in Tab. 5.
>
> Thank you and we are happy to discuss further to clarify any method detail or result including these new results.

---

> > ### Author Rebuttal · Reviewer_uxP6 · 2026-04-06
> >
> > Thanks for the response. All my concerns are addressed and I think the method provides insights on self-supervised model soups. I will raise my score to 4 in this case

---

> > > ### Author Response · Authors · 2026-04-06
> > >
> > > Thank you for your time and attention in reviewing this work and engaging in the discussion.

---

### Decision · Program_Chairs · 2026-04-30

**Decision:**

Accept (spotlight)

**Comment:**

All the reviewers found the empirical study thorough and considered the main finding valuable. Particularly, models inter-trained with different SSL objectives can still be effectively souped, yielding gains in transfer and robustness without relying on extra labeled data. The main concerns pre-rebuttal were about novelty because of its similarity to Model Ratatouille and the fairness of comparison, especially against supervised soups and related baselines. The rebuttal addressed these concerns well. In particular, the authors added controlled comparisons to Ratatouille and supervised soups, and better explained the design choices and terminology.

All the reviewers appreciated the authors' responses, as reflected by the raised scores. I agree with the reviews and believe that this paper would be a solid contribution to the ICML community.